# Ongoing uncoordinated anthropogenic emission abatement promotes atmospheric new particle growth in a Chinese megacity

Lizi Tang[1,4], Zeyu Feng[1,4], Dongjie Shang[1], Linghan Zeng [1], Zhijun Wu[1,2], Hui Wang[3], Shiyi Chen[1], Xin Li [1,2], Limin Zeng[1,2], Jianlin Hu [2] & Min Hu [1,2] ✉

Atmospheric new particle growth in diameter is the crucial process determining air quality effects raised by secondary aerosols. However, uncertain mechanisms and long-term trends of new particle growth limit the assessments of urban air quality evolution. Here we report an increasing trend of new particle growth rate in responds to anthropogenic emission abatement in urban Beijing during autumn from 2017 to 2021. Oxygenated organic vapors is the key compounds driving this variation of growth rate. While the anthropogenic volatile organic precursor abatement has decreased their total concentrations, the concurrent $NO_x$ abatement has increased the fractions thus the concentrations of the low-volatility condensable parts, which are the most relevant contributors for growth. The coeffect of anthropogenic abatement on the volatility distribution of oxygenated organic vapors is the mystery of the increasing growth rate. Our findings highlight the importance of coordinated anthropogenic emission controls on air quality improvement.

Atmospheric new particle formation (NPF) involves gas-phase precursor clustering and subsequent growth of newly-formed particles. The large number of particles generated by NPF contribute over 50% of global cloud condensation nuclei[1–3], and serve as substantial seeds for haze formation in urban atmospheres[4–7]. The new particle growth rate (GR) determines the survival probability of newly-formed particles to larger sizes in the presence of pre-existing aerosols[8]. Enhanced GR could significantly accelerate haze development[5,6]. Accordingly, understanding the characteristics and mechanisms of new particle growth in urban atmospheres is crucial for haze mitigation and policy development.

New particle growth is a complex process involving multiple vapors. Gaseous sulfuric acid (SA) is important for initial growth (<3 nm), while condensation of oxygenated organic molecules (OOMs) has been proposed to dominate subsequent growth into sub-micro particles[9–11]. In urban atmospheres, OOMs primarily form through multi-generation oxidation and auto-oxidation of anthropogenic volatile organic compounds (AVOCs, including aromatic and aliphatic VOCs)[12]. However, high $NO_x$ levels strongly perturb the oxidation process by terminating the $RO_2$ radical auto-oxidation, consequently reducing the oxidation degree and increasing the volatility of OOMs[10,12]. With the progressive implementation of the Air Pollution Prevention and Control Action Plan (2013–2017) and the Three-year Blue-sky Action Plan (2018–2020) in China[13], substantial reductions in AVOC and $NO_x$ emissions have potentially altered OOM formation process, which may influence long-term trends of GR. Nevertheless, our understanding on the long-term evolution of new particle growth and its governing mechanisms under continuous emission controls remains incomplete. Observational data revealed that GR of particles in 3–25 nm exhibited no expected decline during winter from 2013 to 2019 in Beijing, despite continuous reduction in gaseous precursors[14]. Some studies suggest that the stable GR may be attributed to $NO_x$ reduction promoting the formation of low-volatility condensable OOMs that are directly relevant for new particle growth, thus

[1]State Key Laboratory of Regional Environment and Sustainability, International Joint Laboratory of Regional Pollution Control, Ministry of Education (IJRC), College of Environmental Sciences and Engineering, Peking University, Beijing, China. [2]Collaborative Innovation Center of Atmospheric Environment and Equipment Technology, Nanjing University of Information Science and Technology, Nanjing, China. [3]Institute of Climate and Energy Systems, Troposphere, ICE-3, Forschungszentrum Jülich, Jülich, Germany. [4]These authors contributed equally: Lizi Tang, Zeyu Feng. ✉e-mail: minhu@pku.edu.cn

potentially offset the suppression effects from precursor source controls[9,14,15]. However, this hypothesis lacks sufficient evidence due to the limitation of long-term OOM measurements, despite existing comprehensive precursor concentration data (e.g., AVOCs and NO$_x$). Existing extended OOM observations have mainly focused on seasonal variations or GR correlations, leaving their interannual evolution and potential impact on new particle growth during ongoing air quality improvement poorly understood[9,16].

In this work, we conducted comprehensive measurements in urban Beijing, a megacity in China, to investigate the long-term trends and mechanisms of new particle growth. Beijing is a typical city to study new particle growth under anthropogenic mitigation because of its frequent NPF processes and long-term emission control measures[14,17]. We used ambient observations to discover the GR trends during autumn from 2017 to 2021. We identified the effects of AVOCs and NO$_x$ on OOM formation through Nitrate-CI-APi-TOF measurements during autumn 2021 combined with comparative analyses with previous measurements. Importantly, we validated the critical role of concurrent precursor (AVOCs and NO$_x$) emission abatement in the long-term variations of both condensable OOM concentrations and GR, using a parameterization scheme derived from the measurements during autumn 2021. Our results underscore the significant impact of anthropogenic emission abatement on new particle growth, demonstrating that coordinated emission controls are crucial for effectively reducing GR and improving air quality.

## Results

### Long-term trends of growth rate and atmospheric precursor concentrations

Particle number size distribution (PNSD, 1.5–700 nm) and trace gases (NO$_x$, SO$_2$, O$_3$, and AVOCs) were measured in urban Beijing during autumn (September to November) from 2017 to 2021, when haze episodes—particularly those triggered by NPF—were most frequent (Supplementary Fig. 3). A total of 105 NPF events were identified, and size-resolved GR were calculated using the mode-fitting method. Here, three size bins (1.5–3 nm, 3–7 nm, and 7–25 nm, corresponding to GR$_{1.5-3}$, GR$_{3-7}$, and GR$_{7-25}$) were selected for GR analysis because: (1) new particles typically reached ~25 nm by the end of growth process, and (2) OOMs involved in growth process exhibit different volatility

thresholds in these three size bins. Although we tested finer subdivisions (7–15 nm and 15–25 nm), the volatility thresholds of OOMs governing particle growth were aligned with those of 7–25 nm bin (Supplementary Fig. 4). Meteorological conditions were stable in 2017–2021 (Supplementary Fig. 5), therefore, their impact on atmospheric pollutant concentrations was negligible. Thus, the observed GR and trace gas variations primarily reflected anthropogenic emission and atmospheric chemistry changes.

The strict emission controls significantly reduced air pollutant concentrations since 2017 (Fig. 1a), with NO$_x$ decreasing by 50.0% (56.9% in NPF days; Supplementary Fig. 6) and AVOCs decreasing by 48.1% in autumn from 2017 to 2021. AVOCs containing more than 4 carbon atoms (AVOCs$_{nC \geq 4}$) were identified as dominant OOM precursors, as most detected OOMs possessed ≥4 carbon atoms[12]. AVOCs$_{nC \geq 4}$ declined by 57.0% (40.1% in NPF days) in autumn from 2017 to 2021, with similar reduction in aromatic and aliphatic VOCs. For SO$_2$, it has maintained at extremely low levels since 2017 (<2 ppb). O$_x$ (O$_x$ = NO$_2$ + O$_3$), a parameter representing regional-scale atmospheric oxidation capacity[18], showed stable or slightly decreasing trends. The condensational sink (CS), which reflects the scavenging effects of pre-existing particles on condensable vapors, remained nearly constant throughout 2017–2021.

Figure 1b reveals significant increasing trends in size-resolved GR in autumn from 2017 to 2021 (Mann–Kendall test, $p < 0.05$). The average GR$_{3-7}$ and GR$_{7-25}$ increased from 2.2 ± 1.2 and 2.9 ± 1.3 nm h$^{-1}$ in 2017 to 3.2 ± 1.3 and 3.9 ± 1.7 nm h$^{-1}$ in 2021, with annual increasing rate of 0.15–0.38 nm h$^{-1}$ yr$^{-1}$ and 0.12–0.32 nm h$^{-1}$ yr$^{-1}$, respectively. Measurements of PNSD at 1.5–3 nm were only conducted in autumn 2018 and 2021. Higher GR$_{1.5-3}$ were observed in 2021 compared to 2018, although the comparison between the two years cannot be considered representative of a trend. These findings illustrate the enhanced new particle growth in the context of anthropogenic emission reduction.

### Relationship between OOM evolution and GR trends

To understand the cause of the long-term increase in size-resolved GR, we identified the main contributors to new particle growth using Nitrate-CI-APi-TOF measurements during autumn 2021 in urban Beijing. The characteristics of OOMs are shown in Supplementary Fig. 14. A dynamic vapor condensation model was utilized to simulate the vapor

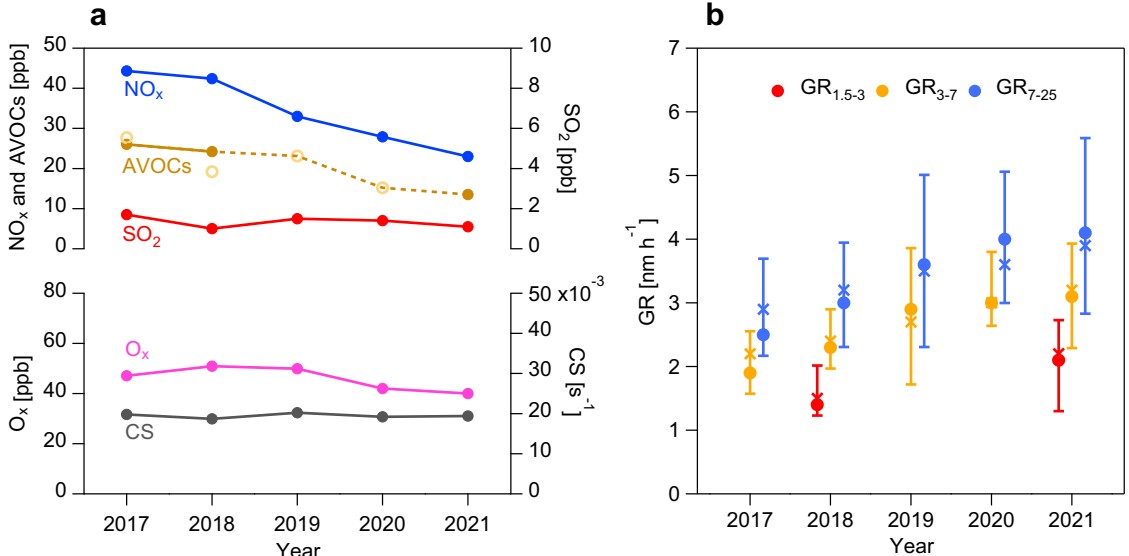

**Fig. 1 | Variations of growth rate (GR), gaseous pollutant concentrations and condensational sink (CS) in autumn from 2017 to 2021. a** Variations of NO$_x$, anthropogenic volatile organic compounds (AVOCs), SO$_2$, O$_x$ (NO$_2$ + O$_3$), and CS. The solid circular markers are values measured in this study, and the hollow circular markers of AVOCs are from Liu et al.[43] as a supplement. **b** Variations of GR of particles in the size ranges of 1.5–3 nm, 3–7 nm, 7–25 nm (GR$_{1.5-3}$, GR$_{3-7}$ and GR$_{7-25}$). The circular and cross markers represent the median and mean values, respectively. The whiskers correspond to the 25th and 75th percentiles.

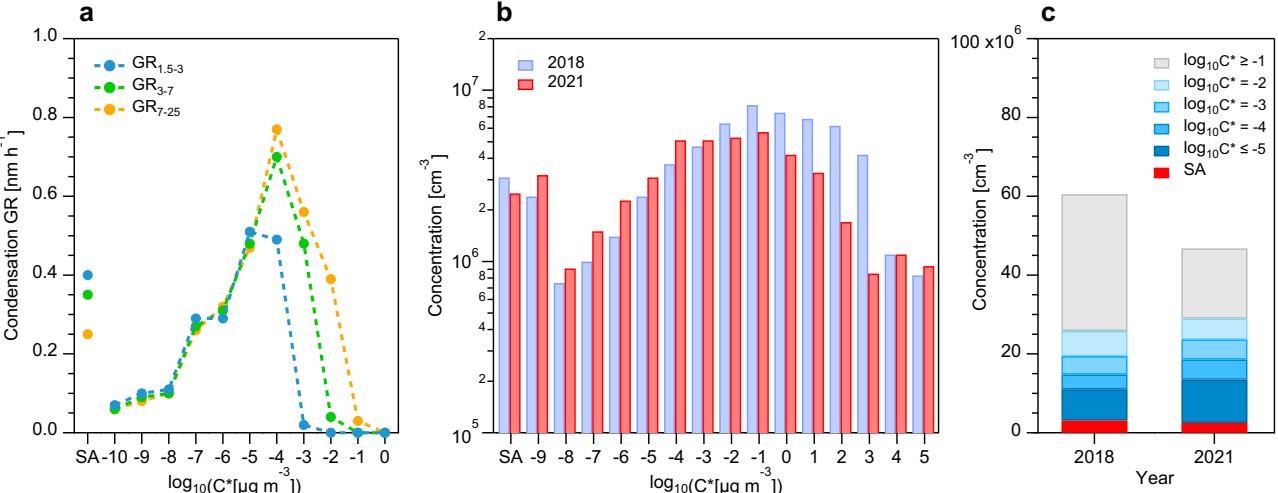

**Fig. 2 | Volatility distribution of oxygenated organic molecules (OOMs) and its link to growth rate (GR). a** Condensation GR contributed by OOMs with different volatility for 1.5–3 nm, 3–7 nm, and 7–25 nm particles. **b** Volatility distribution of sulfuric acid (SA) and OOMs at ambient temperature in new particle formation (NPF) days in autumn 2018 and 2021 in Beijing. Bars corresponding to the locations of −9 and 5 on the horizontal axis are the total concentrations of OOMs with $\log_{10}C^* \le -9$ and $\log_{10}C^* \ge 5$, respectively. **c** Concentrations of SA and OOMs with different volatility in 2018 and 2021. The concentration of SA and the volatility distribution of OOMs in autumn 2018 is from Qiao et al.[20].

condensation involved in the growth process. The simulations revealed that OOMs and SA could explain 80–100% of the observed GR below 25 nm (Supplementary Fig. 7), with negligible coagulation contributions to the observed GR (Supplementary Method 5, Figs. 26 and 27).

OOMs dominated particle growth across all size ranges, contributing an average of 68%, 80%, and 70% to $GR_{1.5-3}$, $GR_{3-7}$, and $GR_{7-25}$, respectively. The oversaturated concentration of a specific OOM determines whether it can condense onto a particle of a specific diameter or not[8]. Due to the Kelvin effect, lower volatility is required for OOMs to participate in the growth process of smaller particles[19]. Based on the measured OOM concentrations in Beijing during autumn 2021, condensation $GR_{1.5-3}$, $GR_{3-7}$, and $GR_{7-25}$ were mainly contributed by OOMs with volatility of $\log_{10}C^* \le -4$, $\log_{10}C^* \le -3$, and $\log_{10}C^* \le -2$, respectively (Fig. 2a). OOMs with $\log_{10}C^* > -2$ could hardly condense onto particles below 25 nm, consistent with another study in 2018 in Beijing[20]. Sensitivity analysis of condensation GR to OOM concentrations and temperature confirmed that these volatility thresholds remained unchanged under typical atmospheric conditions in Beijing (Supplementary Fig. 8). Therefore, the concentrations of OOMs with $\log_{10}C^* \le -4$, $\log_{10}C^* \le -3$, and $\log_{10}C^* \le -2$ largely determined the levels of $GR_{1.5-3}$, $GR_{3-7}$, and $GR_{7-25}$, respectively. Here, we define the size-dependent, growth-relevant OOMs as condensable OOMs.

To distinguish the GR contribution capacity of condensable OOMs with different volatility, we introduced the condensation potential (CP) —the GR contribution per unit concentration of OOMs with specific volatility. As shown in Supplementary Fig. 9, OOMs with lower volatility exhibited higher CP across all size ranges. For example, in 7–25 nm, OOMs with $\log_{10}C^* \le -4$ contributed more to GR per unit concentration than those with $\log_{10}C^* = -4$, $-3$, and $-2$. This phenomenon occurs because CP is essentially determined by the volatility and molar mass of OOMs, while the average molar mass of condensable OOMs remained similar across these volatility ranges[21]. Additionally, lower CP was observed with increasing particle size due to the reduced diameter change per vapor molecule. Owing to non-volatility, SA can overcome the Kelvin effect and effectively contribute to nanoparticle growth, particularly in 1.5–3 nm. However, due to its lower vapor concentrations, lower molar mass, and earlier diurnal evolution peak compared to condensable OOMs (Supplementary Fig. 10), the

contribution of SA to GR was limited, accounting for an average of 32% of $GR_{1.5-3}$ and less than 20% of GR of particles above 3 nm.

Attributed to the stable SA levels in recent years[22], and the dominant role of condensable OOMs in GR as discussed above, the observed GR increase from 2017 to 2021 may have resulted from rising condensable OOM concentrations over this period. Here, we compared the volatility distribution of OOMs for NPF days obtained using our adopted volatility calculation method in autumn 2021 with that reported in autumn 2018 in urban Beijing (Fig. 2b, c)[20]. Although the observation conditions in autumn 2018 and 2021 are not perfectly aligned, their seasonal consistency and temporal overlap permit meaningful comparison. Note that we also used the volatility calculation method from Qiao et al.[20] to calculate the volatility distribution of OOMs in autumn 2021, which was consistent with those obtained using the method adopted in this study, especially for $\log_{10}C^* \le -2$ (Supplementary Fig. 11)[20]. The agreement further suggests that the comparative analysis between the two observations are meaningful. The SA concentrations were comparable between 2018 and 2021. The total OOM concentrations in 2018 were higher than those in 2021, with the difference primarily attributed to compounds in the higher volatility ranges ($\log_{10}C^* > -2$). On the contrary, the higher absolute concentrations of condensable OOMs were observed in 2021, particularly for OOMs with $\log_{10}C^* \le -4$ which have the highest CP. These results suggest that enhanced condensable OOM levels (particularly those with higher CP) have promoted new particle growth since 2017.

## Response of condensable OOMs to emission controls

The increased concentrations of condensable OOMs since 2017 could result from variations in their precursor sources, oxidation process, and condensation sinks. Our analysis shows that ~80% of OOMs formed through the oxidation of AVOCs (including aromatic and aliphatic VOCs, Supplementary Figs. 11 and 12), suggesting that declining AVOC emissions since 2017 likely drive the reduction in OOM concentrations. The response of OOM concentrations to AVOCs will be discussed later based on the intensive campaign in autumn 2021 and comparison with other measurements. Temperature and oxidant concentrations are two essential factors affecting OOM oxidation process[16,23]. Observational results demonstrated that both temperature and $O_x$ (as a proxy for oxidant levels) remained relatively stable both across all

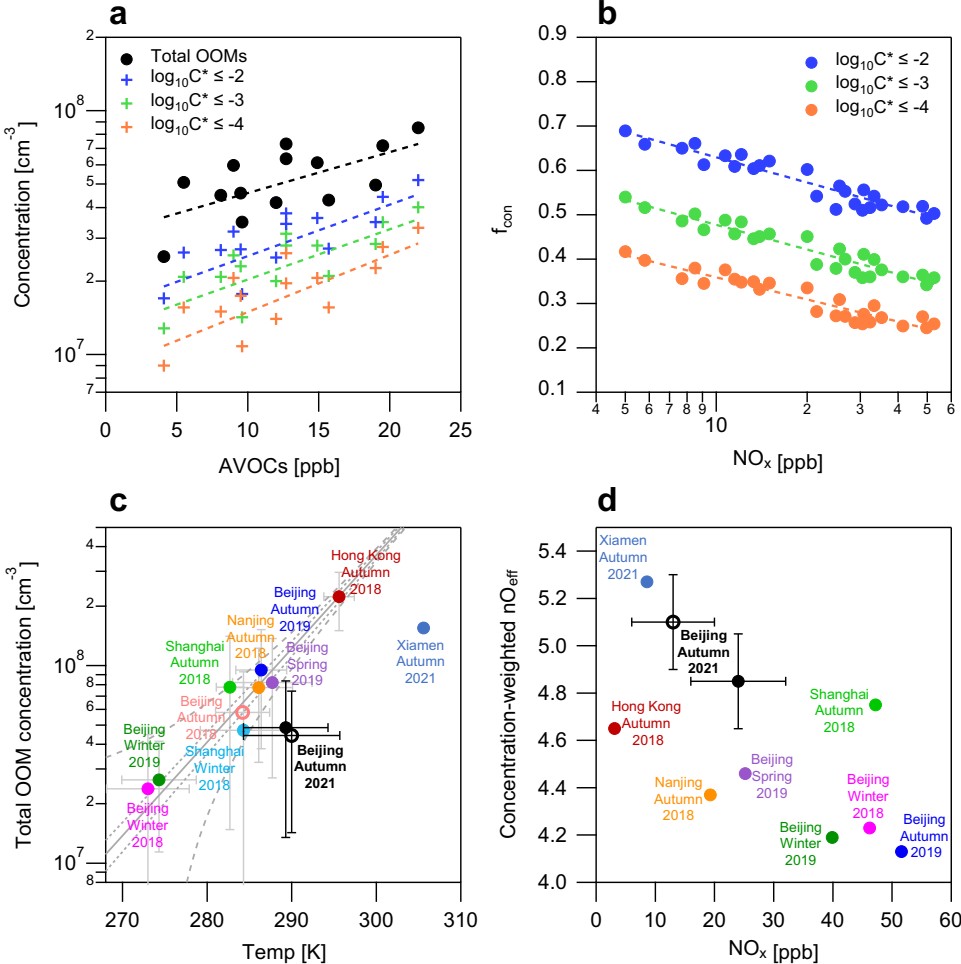

**Fig. 3 | Impact of anthropogenic volatile organic compounds (AVOCs) and $NO_x$ on condensable oxygenated organic molecules (OOMs). a** Relationships between daily average concentrations of OOMs and AVOCs at campaign-average temperature (289 K) during the observation in autumn 2021 in Beijing. The dashed lines represent the linear fitting lines, and the fitting results are shown in "Methods". **b** Relationships between daily average fractions of condensable OOMs ($f_{con}$) and daily average $NO_x$ concentrations at campaign-average temperature (289 K) during the observation in autumn 2021 in Beijing. The dashed lines represent the logarithmic fitting lines, and the fitting results are shown in "Methods". The campaign-average temperature (289 K) was employed to eliminate temperature effects on volatility. **c** Relationships between total OOM concentrations and temperature from different measurements in urban atmospheres in China. Data for spring, autumn and winter 2019 in Beijing are from Guo et al.[16], data for winter 2018 in Beijing, autumn 2018 in Shanghai, Nanjing and Hong Kong are from Nie et al.[12], data

for winter 2018 in Shanghai are from Tian et al.[28], data for autumn 2021 in Xiamen are from Yang et al.[27], and data for NPF days in autumn 2018 in Beijing are from Qiao et al.[20]. The gray lines are the logarithmic fitting curve, 90% confidence and prediction band of data before 2020, respectively. Relationships between total OOM concentrations and temperature in other atmospheres are shown in Supplementary Fig. 22. **d** Relationships between the concentration-weighted effective oxygen number ($nO_{eff}$) of OOMs and $NO_x$ concentrations from different measurements. Data for spring, autumn and winter 2019 in Beijing are from Guo et al.[16], data for winter 2018 in Beijing, autumn 2018 in Shanghai, Nanjing and Hong Kong are from Nie et al.[12], and data for autumn 2021 in Xiamen are from Yang et al.[27]. The filled and hollow circles in (**c**, **d**) are average values from all measurement days and NPF days, respectively. The error bars show standard deviations. Note that the OOM data in Xiamen is anthropogenic OOM data, which can represent the levels and characteristics of the total OOMs.

measurement days and across NPF days since 2017 (Supplementary Fig. 5), indicating that neither temperature nor oxidant concentrations were the key factors controlling the increase in condensable OOM concentrations. Additionally, $NO_x$ can influence the OOM oxidation process by terminating $RO_2$ radical auto-oxidation in urban atmospheres[24]. The sustained $NO_x$ emission reduction since 2017 may have affected OOM composition and condensable OOM concentrations, as will be discussed later. The CS remained stable both across all measurement days and across NPF days since 2017 (Supplementary Fig. 5), suggesting that the increase in condensable OOM concentrations was independent of sink variations.

Our intensive measurement campaign in autumn 2021 reveals significant positive correlations between the concentrations of both total OOMs and condensable OOMs (with different volatility ranges) and precursor AVOC levels (also $AVOCs_{nC \geq 4}$ levels) (Fig. 3a and

Supplementary Fig. 15). These results demonstrate that OOM formation in urban atmospheres is highly sensitive to precursor concentrations, with reduced precursor concentrations directly decreasing OOM production. The fractions of condensable OOMs among total OOMs ($f_{con}$) appears unaffected by AVOC concentrations.

The influence of AVOC levels on OOM concentrations was further corroborated through comparative analyses with other measurements. As shown in Fig. 3c, a significant positive correlation was observed between total OOM concentrations and temperature in urban atmospheric observations across China (Mann–Kendall test, $p < 0.05$). This likely occurs because high temperature (often accompanied by intense solar radiation) increases oxidant concentrations and accelerates $RO_2$ auto-oxidation rates, thereby enhancing OOM production[23,25,26]. Additionally, high temperature may promote elevated biogenic emissions and increase biological OOM concentrations, although this effect is

likely less pronounced in urban atmospheres in China[25]. Notably, the average results from measurements prior to 2020 followed a well-defined OOM-temperature curve ($R^2 > 0.9$). However, results from Beijing and Xiamen during autumn 2021 markedly deviated from this established curve. Under comparable temperatures, OOM concentrations in 2021 were multiple times lower than those in 2019 and earlier years. Potential drivers for the observed deviation include differences in precursor sources or CS. Observation data revealed that the CS levels in Beijing and Xiamen during 2021 were comparable to or lower than those in previous measurements, suggesting that the lower AVOC levels in 2021 may represent the primary explanatory factor. The campaign-average concentrations of $AVOCs_{nC \geq 4}$ in 2021 in Beijing (5.8 ppb) and Xiamen (3.4 ppb) were significantly lower than those in 2019 and earlier years (e.g., 12.0 ppb in Beijing and 12.2 ppb in Shanghai during 2018)[27,28]. In addition, the reduced AVOC precursor levels led to lower OOM concentrations in NPF days than those in all measurement days under comparable temperature in autumn 2021 (Fig. 3c and Supplementary Fig. 10). Based on the observed promoting effects of AVOCs on OOM formation, we conclude that the continued AVOC emission abatement would effectively decrease total OOM concentrations, including condensable OOMs.

The observational results in autumn 2021 reveal that $NO_x$ levels significantly affect the oxidation process of OOMs and the formation of condensable OOMs in urban atmospheres[24]. As shown in Supplementary Fig. 16, higher $NO_x$ concentrations raised the proportion of nitrogen-containing OOMs. In contrast, the fractions of condensable OOMs with $\log_{10}C^* \leq -4$, $\log_{10}C^* \leq -3$, and $\log_{10}C^* \leq -2$ decreased significantly with increasing $NO_x$ levels (Fig. 3b). This occurs because nitrogen-containing OOMs generally exhibit higher volatility than non-nitrogen OOMs[24]. $NO_x$ inhibition on condensable OOM formation was observed for both aromatic-derived and aliphatic-derived OOMs (Aro-OOMs and Ali-OOMs in Supplementary Fig. 17).

The effective oxygen number ($nO_{eff}$, $nO_{eff} = nO - 2 \times nN$) serves as a good parameter for assessing the oxidation degree of OOMs. In this study, we observed a strong positive correlation between concentration-weighted $nO_{eff}$ and $f_{con}$ (Supplementary Fig. 18), suggesting that increased $f_{con}$ was associated with higher oxidation degrees. $nO_{eff}$ can therefore serve as a proxy for $f_{con}$. Figure 3d shows a negative correlation between concentration-weighted $nO_{eff}$ and $NO_x$ across multiple measurements, demonstrating the inhibition effects of $NO_x$ on condensable OOM formation from the perspective of multi-

observation comparison. For measurements conducted in Beijing during autumn, the concentration-weighted $nO_{eff}$ increased from 4.13 in 2019 to 4.85 in 2021 as $NO_x$ decreased by 55%[16]. The increase in $nO_{eff}$ was observed in both Aro-OOMs and Ali-OOMs, with a higher fraction of OOMs exhibiting $nO_{eff}$ above 5 in 2021 compared to that in 2018 and 2019 (Supplementary Fig. 13)[12,16]. For specific molecules, highly abundant Ali-OOMs ($C_6H_{11}NO_6$) and Aro-OOMs ($C_8H_{12}O_5$ and $C_8H_{11}NO_7$) in 2021 contained one additional effective oxygen atom compared to their counterparts in 2019 ($C_6H_{11}NO_5$, $C_8H_{12}O_4$, and $C_8H_{11}NO_6$)[16]. These results confirm that AVOCs have undergone additional oxidation steps under $NO_x$ mitigation, which contributed to the increased $f_{con}$. The OOM sources also affect $f_{con}$ (and $nO_{eff}$), with Aro-OOMs exhibiting higher $f_{con}$ (and $nO_{eff}$) than Ali-OOMs (Supplementary Figs. 13 and 17). This may explain the observed discrepancies in $nO_{eff}$ at comparable $NO_x$ levels across different measurements. For example, measurements from Xiamen and Beijing in 2021, along with Shanghai in 2018, showed a higher fraction of Aro-OOM relative to Ali-OOMs, leading to higher $nO_{eff}$ compared to other measurements at comparable $NO_x$ levels. Additionally, oxidant levels may also influence both $f_{con}$ and $O_{eff}$. Collectively, our analysis demonstrates that the ongoing $NO_x$ emission abatement would promote the formation of highly oxidized OOMs and increase $f_{con}$.

Based on the preceding discussion, the concurrent decline in AVOCs and $NO_x$ since 2017 has yielded two competing effects: (1) an overall decrease in total OOM concentrations, and (2) an increased proportion of condensable OOMs participating in new particle growth. However, the coupled effects from AVOC and $NO_x$ reduction and their roles in driving the long-term increase in condensable OOM concentrations and GR remain unclear. To address these questions, we developed a simplified parameterization scheme based on the OOM observation in autumn 2021, specifically quantifying the response of condensable OOM concentrations and GR to anthropogenic emission abatement during autumn from 2017 to 2021. The core components of this scheme comprise: (1) the relationships between AVOC concentrations and total OOM concentrations, and (2) the relationships between $NO_x$ concentrations and $f_{con}$, both derived from observational data fitting during autumn 2021. The derived condensable OOM concentrations for given AVOC and $NO_x$ concentrations are then converted to GR using the CP (see "Methods" for details). The simulation results under different AVOC and $NO_x$ conditions are presented in Fig. 4 and Supplementary Fig. 19.

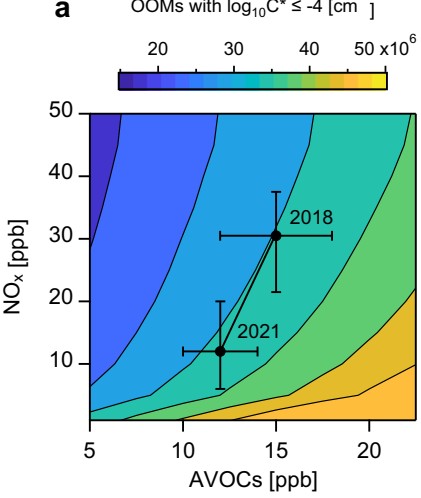
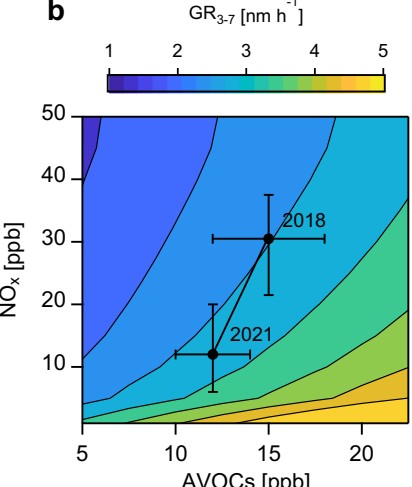
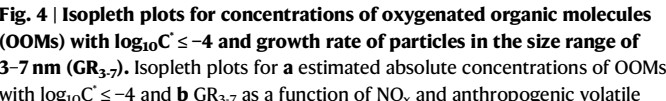

**Fig. 4 | Isopleth plots for concentrations of oxygenated organic molecules (OOMs) with $\log_{10}C^* \leq -4$ and growth rate of particles in the size range of 3–7 nm ($GR_{3-7}$).** Isopleth plots for **a** estimated absolute concentrations of OOMs with $\log_{10}C^* \leq -4$ and **b** $GR_{3-7}$ as a function of $NO_x$ and anthropogenic volatile organic compound (AVOC) concentrations. Black filled circles represent the daily average levels of AVOCs and $NO_x$ measured in NPF days in autumn 2018 and 2021 in Beijing. The whiskers correspond to the 25th and 75th percentiles of $NO_x$ and AVOC concentrations.

Notably, the parameterization scheme successfully reproduced the observations during NPF days in autumn 2018, with discrepancies between measured and simulated condensable OOM concentrations and size-resolved GR below 25% (Supplementary Fig. 20). The successful application of the parameterization scheme (developed using the dataset in 2021) to the dataset of 2018 provides critical evidence that the observed increase in condensable OOMs and GR during 2017–2021 was primarily driven by the predominance of $NO_x$ abatement's enhancing effects over AVOC abatement's inhibitory effects. Note that this scheme does not explicitly consider the effects of temperature, oxidant concentrations, or CS, as these parameters remained relatively stable throughout 2017–2021. The OOM sources were treated as unchanged in the scheme, which may introduce simulation biases. Furthermore, the limited availability of OOM data in 2021 could also introduce additional uncertainties. We emphasize that this parameterization scheme was derived from measurements in autumn 2021, and its validity outside the observed parameter space remains unconfirmed, requiring further verification in future research.

## Discussion

In this study, we observed increasing trends of size-resolved GR in urban Beijing during autumn from 2017 to 2021. Through a comprehensive observation in autumn 2021, we established quantitative relationships between GR, OOMs, and the precursors (AVOCs and $NO_x$). These results identify the uncoordinated emission abatement of AVOCs and $NO_x$ as the primary driver of the annual GR increase. Abatement of AVOCs reduced the concentrations of total OOMs, however, the concurrent reduction of $NO_x$ raised the fractions of condensable OOMs. As a result, these complex interactions elevated the concentrations of highly oxidized condensable OOMs, which in turn enhanced GR.

The impact of $NO_x$ on both OOMs and GR has been investigated in earlier studies. Li et al.[9] demonstrated that high $NO_x$ levels lead to insufficient condensable OOM concentrations and thus limited particle growth in urban areas, which aligns well with the principal conclusions of this study. Furthermore, recent studies have shown that reduced NO levels can activate nocturnal nitrogen chemistry and provide substantial sources of chlorine radicals for daytime reactions[29]. This process can significantly promote VOC oxidation and subsequent OOM formation. However, while this enhanced atmospheric oxidation capacity may play a critical role under polluted conditions, its significance in new particle growth under clean conditions may be limited.

Previous studies have demonstrated that elevated GR exacerbates haze formation and severity, highlighting that reducing GR is critical for pollution mitigation[5,6]. Our analysis reveals that prioritizing AVOC controls over $NO_x$ may more effectively suppress GR. The source-segregated OOM contribution to GR serves as a vital basis for targeted AVOC controls. Simulations show that anthropogenic OOMs dominated the total OOM contribution to GR in all size ranges (~88%; Supplementary Fig. 21), with Aro-OOMs being the predominant contributors (~72% of total OOMs) due to their high concentrations and $f_{con}$. Ali-OOMs represent the second largest source (~16%), while biogenic OOMs exhibit minimal contributions to GR. These findings suggest that prioritizing controls of aromatic VOCs could be an effective strategy for GR reduction and air quality improvement.

Beyond direct effects on OOM formation, precursor VOCs and $NO_x$ may also indirectly influence GR by altering oxidant levels (e.g., $O_3$ and chlorine radicals), which should be considered in future research. Furthermore, iodine oxoacid and SA have been proven to contribute to particle growth (particularly for sub-3 nm particles) in polluted urban environments[30], making it another crucial consideration for future GR mitigation strategies. These findings emphasize the need for comprehensive, long-term GR, precursor, and OOM observations to fully elucidate the dynamic evolution and fundamental mechanisms of atmospheric new particle growth, which are critical for developing efficient pollution control strategies.

## Methods

### Overview of the measurements

The measurements were conducted at the Peking University Urban Atmosphere Environment MonitoRing Station (PKUERS) located in the campus of Peking University (39°59′21″N, 116°18′25″E) in the northwest of urban Beijing[31]. The station represents a typical urban area with large amounts of anthropogenic emissions. The measurements of particle number size distribution (PNSD), trace gases (including $NO_x$, $SO_2$, and $O_3$) and meteorological parameters were performed in autumn (September to November) from 2017 to 2021. A total of 105 NPF events were identified. The criteria for a typical NPF event and the calculation of GR based on PNSD are shown in Supplementary Method 1 and Table 1. VOCs were measured in autumn 2017, 2018, and 2021. The intense campaign including measurements of OOMs and SA was conducted from 18 September to 31 October 2021, including 13 NPF events. The photolysis rates, i.e., $J(O^1D)$, were measured during the intense campaign to provide comprehensive information on OOM formation. Details of the measurement equipment are summarized in Supplementary Method 2 and Table 2.

### Measurement of sulfuric acid and OOMs

OOMs and SA were measured using the Nitrate ion based Chemical Ionization with the Atmospheric Pressure interface Time-Of-Flight mass spectrometer (Nitrate-CI-APi-TOF, Aerodyne Research Inc., USA). The long-TOF (LTOF) detector has the m/z resolution of ~7000 Th/Th. Ambient air at a flow rate of 10 LPM and sheath air at a flow rate of 20 LPM are drawn into the ionization source. The sampling line is a straight Teflon tube with a length of 0.8 m and a diameter of 1/2 in. A mixture of 2.5 ml min⁻¹ ultrahigh purity nitrogen flow containing nitric acid is flushed over the ionizer to generate nitrate as reagent ions. The ions are guided into the sample flow with an electric field, where they react with SA and OOMs. The examples for peak identification are shown in Supplementary Fig. 28. The SA concentration (molec cm⁻³) measured with the Nitrate-CI-APi-TOF are calculated from the measured ion signals according to:

$$[H_2SO_4] = C \times \ln\left(1 + \frac{[HSO_4^-] + [HSO_4^- \cdot HNO_3]}{[NO_3^-] + [NO_3^- \cdot HNO_3] + [NO_3^- \cdot (HNO_3)_2]}\right) \quad (1)$$

Here, $C$ is the calibration coefficient of SA. It is obtained by introducing a known amount of gaseous SA produced by the reaction of $SO_2$ and OH radicals from UV photolysis of water vapor, following the method proposed by Kürten et al.[32]. The calibration was conducted with the sampling tube equipped, so the diffusion loss was contained in the calibration results. The calibration curve of SA is shown in Supplementary Fig. 1. The calibration coefficient of SA is $1.23 \times 10^{10}$ cm⁻³ in autumn 2021. The zeroing background concentration of SA is $5.0 \times 10^4$ cm⁻³ (1 min integration). The limit of detection (LOD) for SA is $9.0 \times 10^4$ cm⁻³ and it is defined as three times the standard deviation of the background.

The concentration of OOMs at $m/z = i$ are calculated as follows:

$$[OOM_i] = C_i \times T_i \times \ln\left(1 + \frac{[OOM_i \cdot NO_3^-]}{[NO_3^-] + [NO_3^- \cdot HNO_3] + [NO_3^- \cdot (HNO_3)_2]}\right) \quad (2)$$

Here, $C_i$ is the calibration coefficient of $OOM_i$. Due to the lack of the structural information of these detected OOMs, direct calibration using an OOM standard is impossible as yet. We assumed that $C_i$ equals $C$, that is, all detected OOMs have the same ionization efficiency as SA

and the $(OOM_i \cdot NO_3^-)$ clusters are very stable, with no dissociation during their residence time of detection[12]. $T_i$ is the mass-dependent relative transmission efficiency, which was obtained based on reagent ion depletion method by introducing a series of perfluorinated acid vapors of different molecular masses to the instrument[33]. The relative transmission efficiency curve was shown in Supplementary Fig. 2. The uncertainty of OOM concentrations are mainly caused by several sources: the uncertainty in the $C$ of SA, which is treated as 33% in this study[32]. Second, the uncertainty raised from adopting $C$ of SA for $C_i$ of OOMs, which is estimated as ±50%[34]. Third, uncertainties may arise from assuming equal ionization efficiency for all OOMs, which might result an underestimation of a factor of 4[19]. In all, the calibrations used in our study provide a lower limit estimate of OOM concentrations, representing the optimized option currently available based on our knowledge.

## Sources of OOMs
In urban atmospheres, OOMs are mainly formed from the oxidation of AVOCs and biological VOCs[12,35]. Based on the current knowledge of VOC oxidation, a recently developed workflow attributes OOMs to four key precursors, including aromatic VOCs, aliphatic VOCs, monoterpene, and isoprene[12,16]. Briefly, the OOMs sources are identified based on the molecular compositions and the up-to-date knowledge of gaseous OOM formation chemistry. The detailed workflow is introduced in Supplementary Method 3.

## Volatility of OOMs
Due to the fact that Nitrate-CI-APi-TOF can only obtain the molecular formula of OOMs without structural information, parameterization methods are usually used to estimate the volatility of OOMs[36]. For OOMs formed from aliphatic, aromatic VOCs and isoprene, they mainly contain hydroxyl, nitrate, and carbonyl groups and very limited hydroperoxide groups[37–39]. The saturation mass concentrations of aromatic, aliphatic and isoprene OOMs at 300 K can be given as follows[36]:

$$\log_{10} C^*(300\,\mathrm{K}) = (25 - nC) \cdot bC - (nO - 2nN) \cdot bO$$
$$- \left[\frac{nC \cdot (nO - 2nN)}{nC + (nO - 2nN)}\right] \cdot bCO \tag{3}$$

where $nC$, $nO$, and $nN$ are the number of carbon, oxygen, and nitrogen in each molecule. $bC = 0.475$, $bO = 2.3$, and $bCO = -0.3$.

For OOMs formed from monoterpene, they mainly contain hydroperoxide groups, which are generated via autoxidation[19,40]. The saturation mass concentrations of these OOMs at 300 K can be given as follows[40]:

$$\log_{10} C^*(300\,\mathrm{K}) = (25 - nC) \cdot bC - (nO - 3nN) \cdot bO - 2 \cdot$$
$$\left[\frac{nC \cdot (nO - 3nN)}{nC + (nO - 3nN)}\right] \cdot bCO - nN \cdot bN \tag{4}$$

where $nC$, $nO$, and $nN$ are the number of carbon, oxygen, and nitrogen in each molecule, respectively. $bC = 0.475$, $bO = 0.2$, $bCO = 0.9$, and $bN = 2.5$.

This is an updated version of the parametrization from Donahue et al.[36]. The modification reflects the extensive presence of hydroperoxide functional groups in OOMs formed via autoxidation, which have a much smaller effect on volatility than the hydroxyl and carbonyl functional groups.

The saturation mass concentrations of OOMs at ambient temperature $T$ is obtained by ref. 41:

$$\log_{10} C^*(T) = \log_{10} C^*(300\,\mathrm{K}) + \frac{\Delta H_{\mathrm{vap}}}{R\ln(10)}\left(\frac{1}{300} - \frac{1}{T}\right) \tag{5}$$

The evaporation enthalpy $\Delta H_{vap}$ is obtained by:

$$\Delta H_{\mathrm{vap}}\left[\mathrm{kJ\,mol^{-1}}\right] = -5.7\log_{10} C^*(300\,\mathrm{K}) + 129 \tag{6}$$

In this study, we first distinguished the sources of OOMs and then calculated the volatility of OOMs from each source using corresponding volatility estimation parameterization. Finally, the OOMs were categorized into volatility bins using a volatility basis set (VBS). SA is considered non-volatile. We also tested the volatility parameterization methods reported in Qiao et al.[20], and the results from both methods are similar, especially in $\log_{10} C^* \leq -2$ ranges (Supplementary Fig. 11).

## Vapor condensation growth modeling
Here, we used a dynamic vapor condensation model to simulate the process of new particle growth driven by the condensation of OOMs and SA[19,21]. The model uses the measured VBS distribution as input to model the mass transfer from the gas to the particle phase. Each VBS bin is regarded as a single surrogate species with the average molar mass and concentration. The detailed information about vapor condensation growth modeling is present in Supplementary Method 4.

## Analysis on the combined effects of AVOCs and NO$_x$ on condensable OOMs and GR
One parameterization scheme establishes the coupled effects of AVOCs and NO$_x$ on condensable OOMs and GR during autumn from 2017 to 2021 in Beijing through: (1) the relationships between AVOC concentrations and total OOM concentrations, (2) the relationships between NO$_x$ concentrations and f$_{con}$, and (3) CP derived from the relationships between condensation GR and condensable OOM concentrations. These relationships are all derived from observational data fitting in autumn 2021.

The relationships between AVOC concentrations and total OOM concentrations were determined through linear regression (Fig. 3a, $y = 2.0 \times 10^6 \times x + 2.8 \times 10^7$, $R^2 = 0.44$), as oxidation product concentrations typically exhibit linear positive correlations with precursor levels. The relationships between NO$_x$ concentrations and f$_{con}$ was established using logarithmic fitting (Fig. 3b). The relationships between NO$_x$ and f$_{con}$ for OOMs with $\log_{10} C^* \leq -2$, $\leq -3$, and $\leq -4$ are $y = -0.19 \times \log_{10}(x) + 0.82$, $R^2 = 0.90$; $y = -0.19 \times \log_{10}(x) + 0.66$, $R^2 = 0.91$; and $y = -0.17 \times \log_{10}(x) + 0.52$, $R^2 = 0.88$, respectively. The logarithmic fitting approach was adopted based on: (1) the observational data exhibit a distinct logarithmic relationship, and (2) previous laboratory experiments and modeling simulations of OOM formation from biogenic and anthropogenic VOC oxidation consistently support this logarithmic dependence (Supplementary Fig. 23). By multiplying the total OOM concentrations and f$_{con}$ under different conditions of AVOCs and NO$_x$, the condensable OOM concentrations with different volatility can be obtained. The above relationships are established on the basis of daily average data in autumn 2021, and thus the calculated condensable OOM concentrations represent the daily average values.

We employed CP of OOMs with different volatility to convert daily average condensable OOM concentrations into size-resolved GR. The CP was obtained by linear fitting the simulated size-resolved condensation GR with the corresponding daily average condensable OOM concentrations (Supplementary Fig. 24), which differs from the CP obtained based on OOM concentrations during the growth period in Supplementary Fig. 9. Since condensation GR is primarily determined by the condensable OOM concentrations during the growth period rather than the daily average concentrations, the linear fitting between the condensation GR and daily average OOM concentrations typically exhibits a y-intercept. When converting GR, it is necessary to consider not only the slope but also the influence of the intercept.

The steps for converting the concentrations of condensable OOM with different volatility into size-resolved GR are as follows: First, we

considered that SA concentrations were constant in recent years, contributing 0.4, 0.35 and 0.25 nm h$^{-1}$ for GR$_{1.5-3}$, GR$_{3-7}$ and GR$_{7-25}$, respectively (Fig. 2a). Secondly, the concentrations of OOMs with log$_{10}$C$^* \leq -4$ were substituted into their respective fitting equations with GR$_{1.5-3}$, GR$_{3-7}$ and GR$_{7-25}$ to obtain the contribution from OOMs with log$_{10}$C$^* \leq -4$ to each size-resolved GR. The contributions from OOMs with log$_{10}$C$^* \leq -4$ were then summed with those from SA to obtain the final GR$_{1.5-3}$. Next, the concentrations of OOMs with log$_{10}$C$^* = -3$ were substituted into their respective fitting equations with GR$_{3-7}$ and GR$_{7-25}$ to obtain the contribution from OOMs with log$_{10}$C$^* = -3$ to each size-resolved GR. The contributions from OOMs with log$_{10}$C$^* \leq -4$ and log$_{10}$C$^* = -3$ were then summed with those from SA to obtain the final GR$_{3-7}$. Finally, the concentrations of OOMs with log$_{10}$C$^* = -2$ were substituted into their respective fitting equations with GR$_{7-25}$ to obtain the contribution from OOMs with log$_{10}$C$^* = -2$ to GR$_{7-25}$. The contributions from OOMs with log$_{10}$C$^* \leq -4$, log$_{10}$C$^* = -3$, and log$_{10}$C$^* = -2$ were then summed with those from SA to obtain the final GR$_{7-25}$.

Figure 4 and Supplementary Fig. 19 present the simulated concentrations of condensable OOMs with different volatility and size-resolved GR under varying AVOC and NO$_x$ conditions. The simulation results for NPF days during autumn 2018 show good agreement with observed values, with all deviations below 25% for both condensable OOM concentrations and size-resolved GR (Supplementary Fig. 20). This agreement clearly demonstrates that the combined effects of AVOCs and NO$_x$ abatement is the cause of long-term variation of GR.

## Data availability

The data that support the main findings of this study have been deposited in Zenodo (https://doi.org/10.5281/zenodo.15680385)[42]. Source data are provided with this paper.

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

## Acknowledgements

Financial support from the National Key Research and Development Program of China (2022YFC3701000, task1, Author M.H.), the National Natural Science Foundation of China-Creative Research Group Fund (22221004, Author M.H., Z.W.), the Youth Fund of the National Natural Science Foundation of China (42305103, Author D.S.), the National Key Research and Development Program of China (2022YFE0135000, Author D.S.). The authors gratefully acknowledge Douglas R. Worsnop for providing drawing and analysis suggestions, Qi Chen, Xi Cheng, Yin Yu, and Song Guo for providing support on Nitrate CI-APi-TOF measurement and calibration, Haoning Chang for providing support on English expression check, Xingang Liu, Yafei Liu, Jingkun Jiang, Xiaohui Qiao, Aijun Ding, Wei Nie, Chao Yan, Yishuo Guo, Jinsheng Chen, and Chen Yang for providing previous measurement data cited in this study.

## Author contributions

L.T., Z.F., and M.H. designed this work. L.T., Z.F., D.S., L.H.Z., Z.W., H.W., S.C., X.L., L.M.Z., and J.H. collected and analyzed the measurement data. L.T. and Z.F. constructed the model. L.T., Z.F., M.H. and L.H.Z. edited the manuscript. All authors have read and agreed to the published version of the manuscript.

## Competing interests
The authors declare no competing interests.
