## [Transparent Peer Review file · Nature Communications]

Ongoing uncoordinated anthropogenic emission abatement promotes atmospheric new particle growth in a Chinese megacity

Corresponding Author: Professor Min Hu

Version 0:

Reviewer comments:

Reviewer #1

(Remarks to the Author)

This manuscript analyses several years of PNSD measurements, highlighting increased particle growth rates, and includes a short field campaign measuring OOM concentrations. These data are extremely valuable and present a very interesting hypothesis, especially given the mounting evidence that particles from NPF can contribute to urban haze. While the initial sections are well written, featuring clear arguments and thorough data analysis, the latter part of the manuscript significantly declines in quality. Despite the impressive presentation and effort in the data calculations, the back end suffers from questionable analysis and interpretation.

Figure 4 and its associated analyses should be removed entirely, as they lack rigour. Similarly, the last two panels of Figure 3 require proper explanation, as their methodology is currently unclear. Once these issues are addressed and the conclusions appropriately revised, it will be at the editor's discretion to determine if the manuscript fits the journal's remit. At present, I believe the scientific quality is insufficient.

General comments:

Unsupported conclusions

This manuscript begins by presenting growth rates from long-term PNSD measurements, showing a clear increase. PNSD was measured continuously from 3 nm upwards and on two occasions from 1.5–3 nm. The authors also include NO₃

- CIMS data collected in autumn 2021

and compare it with data from Qiao et al. (2018). They note that while total OOM concentrations were higher in 2018, the condensable fraction was greater in 2021, suggesting that this increase could explain the rising growth rates (GRs). Figure 3a demonstrates that higher AVOC concentrations correlate with increased OOMs, while Figure 3b shows that high NO_x reduces the condensable fraction of OOMs. These are excellent analyses, with clear and insightful conclusions.

The authors argue that since condensation sink (CS) has been stable in Beijing for years, differences in OOMs are due to variations in NO_x and AVOCs. They present an "OOMtemperature curve" showing OOM concentrations increasing exponentially with temperature,

but this is introduced without sufficient detail or discussion, leaving the argument unclear.

Additionally, they find that mean effective oxygen content of OOMs decreases with increasing NO_x, an interesting result derived from reanalysis of older data. Up to this point—apart from Figure 3c—the manuscript presents strong and commendable science, although I have some reservations about the way the data were averaged for figures 3a,b (see below).

Significant issues arise in the final figure and associated analyses. The CIMS data is extremely limited, covering only two short autumn periods, with logarithmic fits applied to very noisy data

from just one period (2021). These fits, shown in Extended Data Figure 8 (mislabelled in the methods as Extended Data Figure 9), are poorly explained. The caption and legend contradict each other, claiming the red dots represent either daily data or binned data. If binned, the methodology appears arbitrary and unsuitable for meaningful averaging.

The authors then use these questionable fits—based on as few as 14 datapoints, binned into four points to improve the appearance of the fit—to construct an isopleth estimating OOM concentrations and GRs across a wide range of AVOC and NO_x conditions. This is problematic for several reasons:

1. The isopleths imply predictive confidence in conditions (e.g., low NO_x and high AVOC) that are not supported by the data.

2. Despite earlier emphasis on the importance of temperature (Figure 3c) and condensation sink, these factors are omitted from the isopleth analysis.

3. Other critical influences, such as AVOC composition, oxidant concentrations, and RO₂ levels, are ignored, flying in the face of all of the known OOM literature

Finally, the authors choose to present GR_{7–25} in Figure 4b, a diameter range where their simulations fail to match observations (see Figure S3), undermining their conclusions. This final analysis is deeply flawed and detracts significantly from the manuscript's overall quality.

Some dodgy averages

Figures 3a and 3b are based on arbitrary bins used for averaging. Extended Data Figure 8 reveals that the bins for AVOCs contain very few datapoints, whereas those for NO_x include many. This discrepancy undermines the robustness of the analysis, and I also question why (or even how) the authors calculated the 25th and 75th percentiles for the error bars in Figure 3b. The results would be far more convincing if the raw data were presented instead of relying on arbitrary binning. Additionally, the chosen bins produce an alarmingly good fit, which raises concerns about the validity of the approach.

The temperature curve

Figure 3c: The curves presented are not sufficiently explained. It is unclear why Xiamen data were excluded from the fit (is it because they only have AVOCs OOMs? If so, why is it included at all? It is referred to an awful lot. Additionally, existing measurement data from Beck et al. (2021), Brean et al. (2020, 2024), and Kürten et al. (2016) are missing. While some datasets, such as Beck et al. (2021) ($\sim 10^7$ cm⁻³ HOM at ~ 270 K) and Brean et al. (2024) ($\sim 5 \times 10^7$ cm⁻³ at ~ 295 K), align with the curve, others, like Brean et al. (2020) ($\sim 8 \times 10^7$ cm⁻³ at ~ 300 K) and Kürten et al. (2016) ($\sim 3 \times 10^6$ cm⁻³ at ~ 288 K), do not, although my numbers here are based on an eyeballing of their graphs.

This section appears to argue that temperature is the primary determinant of OOM concentration, yet earlier panels demonstrate a strong dependence on precursor concentrations, and earlier text highlights the role of condensation sink. These inconsistencies weaken the argument, especially the final argument of the paper. Furthermore, the references are incorrect, making it unclear where all datapoints originate. If the Xiamen data only report AOOMs, they should not be included in the graph or the discussion.

Writing quality

The overall writing quality of this manuscript is poor, particularly in the latter sections. Numerous small issues are evident, though I have only highlighted some in my specific comments. Key concepts are introduced without adequate explanation—for instance, the “OOMs-temperature curve” is mentioned without first establishing what it represents. Grammar and tense errors are prevalent, such as the use of “OOMs-temperature curve” instead of the correct “OOM-temperature curve.” Additionally, some phrasing choices are perplexing; for example, line 259 states, “Whatever, the increasing trend [...] is obvious,” which is an inappropriate and unclear expression.

As an expert in the field, I found sections such as the discussion of figures 3c and 3d particularly dense and difficult to follow. Similarly, it is unclear how figure 4 was constructed, making it challenging to interpret. These sections would greatly benefit from a rewrite to present the arguments more cohesively, rather than as a collection of disconnected statements. Studies such as (Jimenez et al., 2009), have effectively integrated ambient mass spectral data from many studies and conveyed their findings in a clear and concise manner, which this manuscript could emulate. Furthermore, the figures and captions are often confusing. For example, figure captions, such as Extended Data Figure 8 deviates from the standard format by failing to describe each panel sequentially, making it unnecessarily difficult to understand. The word “fraction” appears on many Y axes also, but what the “fraction” is can sometimes only become apparent in the figure caption. It would be much easier to state “fraction of OOMs containing nitrogen” in extended data fig. 6 for example.

Specific comments:

Line 22: What does “inherent mechanism” mean?

Line 34-35: This ending to the abstract only makes sense if you also believe that NPF is a major contributor to air quality

Line 47: What do you mean by “speed up” haze formation? Do you mean the onset would be faster, or there would be a greater total mass during haze events? Or both?

Line 51: What do you mean by certain period? Do you mean short period?

Line 57: Where AVOCs dominate what? The VOC concentration? The reactive VOCs?

Line 63: What do you mean “the long-term variation”? Do you mean “a long term variation”?

Line 76: I'm not sure what this sentence means “..and the volatility limits of OOMs that are able to condense in these three particle size bins and thus contribute to GR shows distinguishable features”. What are distinguishable features? Also, the figures are referenced out of order.

Line 105 and following: you need to explain what data you used here. You start the prior section saying “Long-term measurements of particle number size distribution (PNSD)...”.

Line 152: Is it EXACTLY the same period? Or is it just two autumn time datasets?

Line 177: Please explain the n_{Oeff} here.

Line 183: Due, rather than duo?

Line 197: This claim is a bit sudden. What is the “OOM-temperature curve”? I understand this in principle, but it is not sufficiently explained.

Fig 3d: Why are certain sites (non-chinese cities) excluded from this analysis?

Line 206: Multi-generation oxidation OR auto-oxidation

Beck, L. J., Sarnela, N., Junninen, H., Hoppe, C. J. M., Garmash, O., Bianchi, F., Riva, M., Rose, C., Peräkylä, O., Wimmer, D., Kausiala, O., Jokinen, T., Ahonen, L., Mikkilä, J., Hakala, J., He, X.-C., Kontkanen, J., Wolf, K. K. E., Cappelletti, D., Mazzola, M., Traversi, R., Petroselli, C., Viola, A. P., Vitale, V., Lange, R., Massling, A., Nøjgaard, J. K., Krejci, R., Karlsson, L., Zieger, P., Jang, S., Lee, K., Vakkari, V., Lampilahti, J., Thakur, R. C., Leino, K., Kangasluoma, J., Duplissy, E.-M., Siivola, E., Marbouti, M., Tham, Y. J., Saiz-Lopez, A., Petäjä, T., Ehn, M., Worsnop, D. R., Skov, H., Kulmala, M., Kerminen, V.-M., and Sipilä, M.: Differing Mechanisms of New Particle Formation at Two Arctic Sites, *Geophysical Research Letters*, 48, e2020GL091334, <https://doi.org/10.1029/2020GL091334>, 2021.

Brean, J., Beddows, D. C. S., Shi, Z., Temime-Roussel, B., Marchand, N., Querol, X., Alastuey, A., Minguillón, M. C., and Harrison, R. M.: Molecular insights into new particle formation in Barcelona, Spain, *Atmospheric Chemistry and Physics*, 20, 10029-10045, 10.5194/acp-20-10029-2020, 2020.

Brean, J., Rowell, A., Beddows, D. C. S., Weinhold, K., Mettke, P., Merkel, M., Tuch, T., Rissanen, M., Maso, M. D., Kumar, A., Barua, S., Iyer, S., Karppinen, A., Wiedensohler, A., Shi, Z., and Harrison, R. M.: Road Traffic Emissions Lead to Much Enhanced New Particle Formation through Increased Growth Rates, *Environmental Science & Technology*, 58, 10664-10674, 10.1021/acs.est.3c10526, 2024.

Jimenez, J. L., Canagaratna, M. R., Donahue, N. M., Prevot, A. S. H., Zhang, Q., Kroll, J. H., DeCarlo, P. F., Allan, J. D., Coe, H., Ng, N. L., Aiken, A. C., Docherty, K. S., Ulbrich, I. M., Grieshop, A. P., Robinson, A. L., Duplissy, J., Smith, J. D., Wilson, K. R., Lanz, V. A., Hueglin, C., Sun, Y. L., Tian, J., Laaksonen, A., Raatikainen, T., Rautiainen, J., Vaattovaara, P., Ehn, M., Kulmala, M., Tomlinson, J. M., Collins, D. R., Cubison, M. J., E., Dunlea, J., Huffman, J. A., Onasch, T. B., Alfarra, M. R., Williams, P. I., Bower, K., Kondo, Y., Schneider, J., Drewnick, F., Borrmann, S., Weimer, S., Demerjian, K., Salcedo, D., Cottrell, L., Griffin, R., Takami, A., Miyoshi, T., Hatakeyama, S., Shimono, A., Sun, J. Y., Zhang, Y. M., Dzepina, K., Kimmel, J. R., Sueper, D., Jayne, J. T., Herndon, S. C., Trimborn, A. M., Williams, L. R., Wood, E. C., Middlebrook, A. M., Kolb, C. E., Baltensperger, U., and Worsnop, D. R.: Evolution of Organic Aerosols in the Atmosphere, *Science*, 326, 1525-1529, doi:10.1126/science.1180353, 2009.

Kürten, A., Bergen, A., Heinritzi, M., Leiminger, M., Lorenz, V., Piel, F., Simon, M., Sitals, R., Wagner, A. C., and Curtius, J.: Observation of new particle formation and measurement of sulfuric acid, ammonia, amines and highly oxidized organic molecules at a rural site in central Germany, *Atmos. Chem. Phys.*, 16, 12793-12813, 10.5194/acp-16-12793-2016, 2016.

Reviewer #3

(Remarks to the Author)

Tang et al. report the long-term trend of size-resolved particle growth rate in urban area of Beijing during the autumns from 2017 to 2021, and explore the driving factor of growth rate variations based on an intensive campaign with the gaseous oxygenated organic molecules observations. These long-term data can be an important reference for future air pollution control and mitigation works in China. The manuscript is well-written and the presentation is clear. However, I have some queries on the novelty and discussion in the manuscript.

I think that the manuscript should be significantly revised to emphasize more on its novelty and include more solid discussion to show the importance of the long-term observations from a broader perspective. For example, the authors stated in Line 65-67 that relevant ambient measurement evidence is still scarce, and the mechanism of new particle growth

and its long-term evolution need in-depth research. I do agree that the long-term observations of GR are important, however, I do not see that the overall results in the current manuscript have addressed this statement well enough. There are already some previous studies that focus on the long-term observations of new particles growth related to the OOMs in urban Beijing (e.g., Li et al., 2022, ES&T; Guo et al., 2023, ACP). Other than different in time period (season), how do the authors distinguish the novelty of this study compared to the previous observations? This should be clarified. Furthermore, it is not mentioned in the manuscript on why only the autumn data were selected for the analysis. Or why it is important to look at the autumn data? Any justification?

In term of mechanism, linking the growth rate with the OOMs evolution and NO_x level is a key point of this study. It is unfortunate that the OOMs data were only available in 2021, and the trend of the OOMs levels were unknown in other years. Although the authors have implied that the AVOCs decrease will cause decrement in total OOMs (and condensable OOMs) and compared the data from previous studies to show the impacts of NO_x to highly oxidized OOMs formation (Fig.3), how sure are they that trend will not deviate as predicted? It is stated that the ongoing abatement in NO_x emission promoted the formation of highly oxidized OOMs and increased fcon (Line 236-237). Despite supported by the calculations, I am not totally convinced on this conclusion by the current data as they could be also influenced by other factors as shown by previous studies. Yan et al. (2022, ACP) found that OOM composition and volatility were insensitive to the large change of atmospheric NO_x concentration in Beijing during the COVID-19 restrictions; instead, the associated high particle growth rates and high OOM concentration during the lockdown period were mostly caused by the enhanced atmospheric oxidative capacity, which linked to the enhanced night-time NO_x chemistry (Yan et al., 2023, NG). Furthermore, Li et al. (2022, EST) exhibits a different view, where they found that the insufficient condensable organic vapours lead to slow growth, which further causes low survival of the newly formed particles in urban environments of Beijing. Did the author compare their results with more recent findings to get a bigger picture on the trend of OOMs? For example, Yuan et al. (2024, ES&T) reported the atmospheric gaseous OOMs in urban Beijing from January 2022 and January 2023. Can these data being used to support the reduction of NO_x leading to OOMs decrease beyond 2021?

I agree that developing effective strategies to reduce GR is important for mitigating aerosol pollution, but the current discussion in the policy implication section is simplistic. It is not sure how the calculation led to the proposal of 1:3 AVOCs and NO_x abatement ratio. I am wondering if the changes of 1:3 to 1:2 will have a significant effect in the OOMs concentration considering all the uncertainties in the calculation (including the uncertainty in AVOCs and NO_x measurements)? Maybe I am wrong, but at least from arrows in the Fig. 4 and EDF 9, they seem not to show significant different (large shift) in concentrations of OOMs to me. As I have mentioned in the comment above, many other factors can affect the OOMs level, but the limited OOMs observation data (only in 2021) has weakened the validity of the conclusion drawn in this section. I think that the effectiveness to reduce GR in urban area is more complicated than just by controlling the AVOC (aromatic VOCs) because besides the VOCs levels, the levels of oxidants may also play important role in the OOMs formation. This has been neglected throughout the discussion in the text. In addition, iodine oxoacids and sulfuric acids have been proven to contribute to the particle formation and particle growth in polluted urban environments (e.g., Zhang et al., 2024, ACP), may further complicate the control measure in reducing particle growth. I would suggest the authors to revise the implication of this study.

Specific comments:

Line 1: The title of the manuscript is overstated to me, suggest to revise the title accordingly. For example, "Urban China" is not suitable here as I think the current data in the manuscript only represent urban Beijing, instead of the general urban areas of China. I did not see sufficient data to support the general urban areas of China.

Line 45: Should be environmental effects.

Line 76: What kind of distinguishable features?

Line 77-78: Please revise the sentence as it is hard to understand how the measured GR and trace gases can directly link to the anthropogenic emission abatement and changing atmospheric chemistry?

Line 87: Please state the method for estimating the condensational sink (CS) data in the methodology or supporting information.

Line 89: Suggest rewording 'in' to 'throughout'.

Line 91: Suggest to include the standard deviation for the averaged value of growth rate.

Line 131: accompanied by?

Line 148: The volatility distribution of OOMs in 2018 were obtained from Qiao et al., are the OOMs detected in Qiao's study similar to the current study? If not, what are the differences?

Line 259: Please change 'Whatever' to a more appropriate word (same for line 289).

Line 304: Should be 'i.e.',

Line 314: For the sulfuric acid calibration, the authors should state the detail procedures of their calibration conducted for this

study instead of just mentioning the work of Kürten et al. A calibration curve of H₂SO₄ should be included to the supporting information as this is important because the concentration of OOMs were also based on this calibration. What is the determined limit of detection and zeroing background signal for H₂SO₄? All these information should also be added.

Line 387: Revise the typo.

Line 662, Extended Data Fig. 1: Why the AVOCs comparison plot (figure b) only included autumn 2018 and 2021 data, while the VOCs were also measured in autumn 2017 (as mentioned in the methodology)? Please justify. Does it still have a decrease trend after including the 2017 data? In addition, it is not mentioned that what kind/specific species of AVOCs were used for the analysis in this study. This information should be added to a table in the supporting information.

Line Extended Data Fig.4: It is not sure why the Hyytiälä (a forest station) data was included for the comparison of OOMs to data from cities in China. It confuses me as the current manuscript is discussing the OOMs in urban China, isn't it?

Supporting Information, Figure S9: Suggest to separate the r² value according the linear lines as the range of 0.59-0.80 is quite big. It is better to show which line has stronger correlation.

References:

- Li et al., Insufficient condensable organic vapors lead to slow growth of new particles in an urban environment, *Environ. Sci. Technol.*, 56, 14, 9936–9946 (2022).
- Guo et al., Measurement report: The 4-year variability and influence of the Winter Olympics and other special events on air quality in urban Beijing during wintertime, *Atmos. Chem. Phys.*, 23, 6663–6690 (2023).
- Yan et al., The effect of COVID-19 restrictions on atmospheric new particle formation in Beijing, *Atmos. Chem. Phys.*, 22, 12207–12220 (2022).
- Yan et al., Increasing contribution of nighttime nitrogen chemistry to wintertime haze formation in Beijing observed during COVID-19 lockdowns, *Nat. Geosci.* 16, 975–981 (2023).
- Yuan et al., Resolving atmospheric oxygenated organic molecules in urban Beijing using online ultrahigh-resolution chemical ionization mass spectrometry, *Environ. Sci. Technol.*, 58, 17777–17785 (2024).
- Zhang et al., Iodine oxoacids and their roles in sub-3 nm particle growth in polluted urban environments, *Atmos. Chem. Phys.*, 24, 1873–1893 (2024).

Version 1:

Reviewer comments:

Reviewer #1

(Remarks to the Author)

The authors have generally done a good job of responding to my comments. The analysis of the HOM data is first class and they've convinced me that the isopleth figure is worth keeping, with one caveat, which is that I think the fact that there is no evidence that their model works outside of the parameter space of their measurements needs to be noted.

Figures one through three present some really nice analysis. I feel like the authors felt some pressure to tie it together into some coherent story and this has led to some obfuscations and exaggerations throughout the manuscript. Similarly, the English throughout is still below a publishable standard. I'll take one snippet (at random) as an example

"Gaseous condensable OOMs can easily condense onto pre-existing particles and be removed due to their low volatility (1). The CS remained stable throughout both the whole days and NPF days since 2017 (2), suggesting that the increase in condensable OOM concentration were independent of CS variations (3). In urban atmosphere, OOM formation primarily related to photochemical oxidation of AVOCs (~80%, Supplementary Fig. S10 and S12). The relatively constant photochemical reactivity (indicated by Ox levels and temperature) during 2017-2021 exerted negligible influence on condensable OOM concentration variations. (4)"

- 1) "condense onto pre-existing particles and be removed" is ambiguous as it could imply that the condensation and the removal are two separate processes. They instead of "condense onto pre-existing particles, resulting in their removal" or something similar
- 2) "whole days" is incorrect, I think they mean "The CS remained stable both across all measurement days and across NPF days".
- 3) Should say "Condensable OOM concentrations"
- 4) Ox levels and temperature do not indicate "photochemical reactivity". Similarly, "2017-2021" is misleading here because it implies that they know how photochemical reactivity affected OOMs from 2017 through 2021. In reality they do not have a measure of photochemical reactivity, and if they did, they only have OOM measurements for a small fraction of the 2017 to 2021 period.

It is outside of the scope of a reviewer's job to highlight and fix every single one of these as they're consistent through the manuscript.

One final comment is that, despite my previous comment, it's still not totally clear to me what data is used to construct which

plot. The new Figure 3a and b should show the same thing (daily averages), yet one has 14 datapoints while the other has 26.

I'd be extremely happy to see a final version of the manuscript where the English has been tied up.

Reviewer #3

(Remarks to the Author)

I have read the authors response to my comments and those of the other reviewer and find that they have done a good job in addressing the comments. I have only two minor suggestions as below.

1) Line 274-276: References should be provided to the works that have proven the iodine oxoacids and sulfuric acids contributions to particle growth.

2) Fig.S1 in the supplement: The authors should describe the H₂O (lpm) in the caption. Non-expert reader will not understand what does H₂O (lpm) represent for and how do they work.

Reply to comments

Manuscript ID: NCOMMS-24-78518-T

Title: "Ongoing anthropogenic emission abatement promotes new particle growth in a Chinese megacity"

Author(s): Lizi Tang, Zeyu Feng, Dongjie Shang, Linghan Zeng, Zhijun Wu, Hui Wang, Shiyi Chen, Xin Li, Limin Zeng, Jianlin Hu, Min Hu

I. Reply to Reviewer 1

Reply to Reviewer 1's overall comments:

This manuscript analyses several years of PNSD measurements, highlighting increased particle growth rates, and includes a short field campaign measuring OOM concentrations. These data are extremely valuable and present a very interesting hypothesis, especially given the mounting evidence that particles from NPF can contribute to urban haze. While the initial sections are well written, featuring clear arguments and thorough data analysis, the latter part of the manuscript significantly declines in quality. Despite the impressive presentation and effort in the data calculations, the back end suffers from questionable analysis and interpretation.

Figure 4 and its associated analyses should be removed entirely, as they lack rigour. Similarly, the last two panels of Figure 3 require proper explanation, as their methodology is currently unclear. Once these issues are addressed and the conclusions appropriately revised, it will be at the editor's discretion to determine if the manuscript fits the journal's remit. At present, I believe the scientific quality is insufficient.

We appreciate the constructive comments from the reviewer on this manuscript. We have answered them point to point in the following paragraphs (the texts italicized are the comments, the texts indented are the responses, and the texts in blue are revised parts in new manuscript). And we have carefully modified the language before submitting. In addition, all changes made are marked in the revised manuscript.

Reply to Reviewer 1's general concerns (4):

1. (1) This manuscript begins by presenting growth rates from long-term PNSD measurements, showing a clear increase. PNSD was measured continuously from 3 nm upwards and on two occasions from 1.5–3

*nm. The authors also include NO₃- CIMS data collected in autumn 2021 and compare it with data from*
*Qiao et al. (2018). They note that while total OOM concentrations were higher in 2018, the condensable*
*fraction was greater in 2021, suggesting that this increase could explain the rising growth rates (GRs).*
*Figure 3a demonstrates that higher AVOC concentrations correlate with increased OOMs, while Figure*
*3b shows that high NO_x reduces the condensable fraction of OOMs. These are excellent analyses, with*
*clear and insightful conclusions.*

*The authors argue that since condensation sink (CS) has been stable in Beijing for years, differences in*
*OOMs are due to variations in NO_x and AVOCs. They present an "OOM- temperature curve" showing*
*OOM concentrations increasing exponentially with temperature, but this is introduced without sufficient*
*detail or discussion, leaving the argument unclear. Additionally, they find that mean effective oxygen*
*content of OOMs decreases with increasing NO_x, an interesting result derived from reanalysis of older data.*
*Up to this point—apart from Figure 3c—the manuscript presents strong and commendable science,*
*although I have some reservations about the way the data were averaged for figures 3a,b (see below).*

We sincerely appreciate the positive assessment of our research. The “OOM-temperature curve” section
has been rewritten for better understand. Our analysis showed a significant positive correlation between
total OOM concentrations and temperature in urban atmospheric observations across China. This is likely
because high temperatures (often accompanied by intense solar radiation) can lead to increased oxidant
concentrations and accelerated RO₂ auto-oxidation rates, resulting in higher OOM production (Bianchi et
al., 2019; Praske et al., 2018; Zheng et al., 2023). Additionally, high temperatures may promote elevated
biogenic emissions and elevate biological OOM concentrations, though this effect is likely less
pronounced in urban atmosphere in China. The complete observational dataset before 2020 consistently
followed a well-defined OOM-temperature curve ($R^2 > 0.9$). However, this study employed the curve
primarily to illustrate the influence of AVOC concentration. Notably, the OOM concentrations in Beijing
and Xiamen in 2021 deviated from the OOM-temperature pattern before 2020. Under comparable
temperature, the OOM concentrations in 2021 showed multiple times lower compared to 2019 and earlier
54 years. This deviation likely stems from the reduced precursor AVOC concentration in 2021, with
55 concentration of AVOC_{SnC ≥ 4} in 2021 in Beijing (5.8 ppb) and Xiamen (3.4 ppb) much lower than those
56 in 2019 and earlier, such as Beijing (13.9 ppb) and Shanghai (12.2 ppb) in 2018. In addition, OOM
concentration in NPF days in 2018 and 2021 were lower than the complete observation under comparable
temperature, primarily due to the lower level of precursor AVOCs. To better highlight the effects of AVOC
emission reductions, Figure 3c has been updated to exclusively display data points from Chinese cities.

Bianchi, F. et al. Highly Oxygenated Organic Molecules (HOM) from Gas-Phase Autoxidation Involving Peroxy

Radicals: A Key Contributor to Atmospheric Aerosol. *Chemical Reviews* 119, 3472-3509,
doi:10.1021/acs.chemrev.8b00395 (2019).

Praske, E. et al. Atmospheric autoxidation is increasingly important in urban and suburban North America. 115, 64-69,
doi:10.1073/pnas.1715540115 (2018).

Zheng, P. et al. Molecular Characterization of Oxygenated Organic Molecules and Their Dominating Roles in Particle
Growth in Hong Kong. *Environmental Science & Technology* 57, 7764-7776, doi:10.1021/acs.est.2c09252 (2023).

We have made correction in the revised manuscript as follows:

“The influence of AVOC level on OOM concentration was further corroborated by comparative analyses with other
measurements. As shown in Fig. 3c, a significant positive correlation was observed between total OOM
concentrations and temperature in different observation in urban atmospheric observations across China (Mann-
Kendall test, $p < 0.05$). This is likely because high temperatures (often accompanied by intense solar radiation) can
lead to increased oxidant concentrations and accelerated RO₂ auto-oxidation rates, resulting in higher OOM
production.²³⁻²⁵ Additionally, high temperatures may promote elevated biogenic emissions and elevate OOM
concentrations, though this effect is likely less pronounced in urban atmosphere of China. Notably, the complete
observational dataset before 2020 consistently followed a well-defined OOM-temperature curve ($R^2 > 0.9$).
However, results from Beijing and Xiamen during the autumn of 2021 markedly deviated from this established
curve. Under comparable temperature, the OOM concentrations in 2021 showed multiple times lower compared to
2019 and earlier years. Since CS remained stable in 2018-2021, this deviation likely resulted from reduced AVOC
precursors in 2021. The concentration of AVOC_{S_nC_{≥4}} in 2021 in Beijing (5.8 ppb) and Xiamen (3.4 ppb) were much
lower than those in 2019 and earlier, such as Beijing (13.9 ppb) and Shanghai (12.2 ppb) in 2018.^{26,27} In addition,
OOM concentration in NPF days in 2018 and 2021 were lower than the complete observation under comparable
temperature, primarily due to the lower level of precursor AVOCs. Based on the observed promoting effect of
AVOCs on OOM formation, we conclude that continued AVOC emission abatement can effectively decrease total
OOM concentrations, including condensable OOMs.”

Fig. 3. Impact of AVOCs and NO_x on condensable OOM concentration. **a** The relationships between daily average concentration of OOMs and AVOCs at campaign-average temperature (289 K) during the observations in 2021. The dashed line represents the linear fitting line, and the fitting result is shown in Methods. **b** The relationships between daily average fraction of condensable OOMs (f_{con}) and NO_x at average temperature (289 K) during the observations in 2021. The dashed lines represent the logarithmic fitting line, and the fitting results is shown in Methods. **c** The relationships between the concentrations of the total OOMs and temperature from different measurements. Data for spring, autumn and winter 2019 in Beijing are from Guo, et al.¹⁹, data for winter 2018 in Beijing, autumn 2018 in Nanjing and Hong Kong are from Nie, et al.¹⁴, data for autumn 2018 in Shanghai are from Tian, et al.²⁷, data for autumn 2021 in Xiamen are from Yang, et al.²⁶, and data for NPF days in autumn 2018 in Beijing are from Qiao, et al.²⁰. The area formed by the gray lines is the logarithmic fitting curve, 95% confidence and prediction band of data from the complete measurements before 2020. **d** The relationships between the average effective oxygen number (nO_{eff_ave}) of OOMs and NO_x from different measurements. Data for spring, autumn and winter 2019 in Beijing are from Guo, et al.¹⁹, data for winter 2018 in Beijing, autumn 2018 in Shanghai, Nanjing and Hong Kong are from Nie, et al.¹⁴, and data for autumn 2021 in Xiamen are from

98 Yang, et al. ²⁶. The filled circles and hollow circles are average values from whole measurement and NPF days, respectively. The
99 gray error bars show standard deviations. Note that the OOMs in Xiamen is AOOMs. Given the dominant role of AOOMs in urban
atmospheres, their concentrations can serve as a reasonable proxy for all OOM levels.

***(2) Significant issues arise in the final figure and associated analyses. The CIMS data is extremely limited,***
***covering only two short autumn periods, with logarithmic fits applied to very noisy data from just one period***
***(2021). These fits, shown in Extended Data Figure 8 (misabeled in the methods as Extended Data Figure***
***9), are poorly explained. The caption and legend contradict each other, claiming the red dots represent***
***either daily data or binned data. If binned, the methodology appears arbitrary and unsuitable for***
***meaningful averaging.***

Thanks for the comment. As mentioned in the comments, the binned methodology is unsuitable for data
analysis. We have revised all analyses and figures using raw daily average values to clearly present our
results. The noisy data for f_{con} (fraction of condensable OOMs) and NO_x in previous version manuscript
resulted from displaying raw 10-minute resolution data, which has been removed in the revised
manuscript as statistical calculations were actually performed using daily averaged results. We sincerely
apologize for the labeling errors in the previous version of the manuscript.

In addition, we conducted the analyses about the effect of AVOCs and NO_x on OOMs using the campaign-
average temperature (289 K) in the revised manuscript to eliminate temperature effects on volatility,
which significantly improves the fitting performance.

We have made correction in the revised manuscript as follows:

Fig. 3. Impact of AVOCs and NO_x on condensable OOM concentration. **a** The relationships between daily average concentration of OOMs and AVOCs at campaign-average temperature (289 K) during the observations in 2021. The dashed line represents the linear fitting line, and the fitting result is shown in Methods. **b** The relationships between daily average fraction of condensable OOMs (f_{con}) and NO_x at average temperature (289 K) during the observations in 2021. The dashed lines represent the logarithmic fitting line, and the fitting results is shown in Methods. **c** The relationships between the concentrations of the total OOMs and temperature from different measurements. Data for spring, autumn and winter 2019 in Beijing are from Guo, et al.¹⁹, data for winter 2018 in Beijing, autumn 2018 in Nanjing and Hong Kong are from Nie, et al.¹⁴, data for autumn 2018 in Shanghai are from Tian, et al.²⁷, data for autumn 2021 in Xiamen are from Yang, et al.²⁶, and data for NPF days in autumn 2018 in Beijing are from Qiao, et al.²⁰. The area formed by the gray lines is the logarithmic fitting curve, 95% confidence and prediction band of data from the complete measurements before 2020. **d** The relationships between the average effective oxygen number (nO_{eff_ave}) of OOMs and NO_x from different measurements. Data for spring, autumn and winter 2019 in Beijing are from Guo, et al.¹⁹, data for winter 2018 in Beijing, autumn 2018 in Shanghai, Nanjing and Hong Kong are from Nie, et al.¹⁴, and data for autumn 2021 in Xiamen are from

131 Yang, et al.²⁶. The filled circles and hollow circles are average values from whole measurement and NPF days, respectively. The
132 gray error bars show standard deviations. Note that the OOMs in Xiamen is AOOMs. Given the dominant role of AOOMs in urban
atmospheres, their concentrations can serve as a reasonable proxy for all OOM levels.

“Based on the discussion above, the concurrent decline of AVOCs and NO_x since 2017 has led to a reduction in
total OOM concentration, and an increased proportion of condensable OOMs participating in new particle growth.
To quantify these coupled effects on condensable OOM concentration and GR during 2017-2021, we developed a
simplified parameterization scheme based on OOM observation in 2021 (see Methods for detailed methodology).

Methods

Analysis on the combined effect of AVOCs and NO_x on condensable OOMs and GR

This parameterization scheme establishes the coupled effects of AVOCs and NO_x on condensable OOMs and GR
in the autumn of Beijing during 2017-2021 through: (1) the relationships between AVOCs and total OOM
concentration, and (2) the relationships between NO_x and f_{con}, both derived from observational data fitting in the
autumn in 2021. The derived condensable OOM concentration is subsequently converted to GR using condensation
potential (CP).

The relationship between AVOCs and total OOM concentration was determined through linear regression (Fig. 3a,
$y = 2.0 \times 10^6 \times x + 2.8 \times 10^7$, $R^2 = 0.45$), as oxidation product concentrations typically exhibit linear positive
correlations with precursor levels. The relationship between NO_x and f_{con} was established using logarithmic fitting
(Fig. 3b). The relationship between NO_x and f_{con} for $\log_{10}(C^*) \leq -4$, $= -3$, $= -2$ are $y = -0.20 \times \log_{10}(x) + 0.84$, $R^2 =$
0.82 ; $y = -0.19 \times \log_{10}(x) + 0.68$, $R^2 = 0.86$; and $y = -0.16 \times \log_{10}(x) + 0.53$, $R^2 = 0.88$, respectively. The logarithmic
fitting approach was adopted based on: (1) the observational data exhibit a distinct logarithmic relationship, and (2)
previous laboratory experiments and modeling simulations of OOM formation from biogenic and anthropogenic
VOC oxidation consistently support this logarithmic dependence (Supplementary Fig. S23). By multiplying the
total OOM concentration and f_{con} under different conditions of AVOCs and NO_x, the condensable OOMs
concentration of various volatilities can be obtained. The condensable OOM concentration is the daily average value.

**Figure S23.** The relationships between daily average fraction of condensable OOMs and NO_x in the observations in 2021.
The dashed lines represent the logarithmic fitting result. Other laboratory experiment (Monoterpene + OH/O_3 + NO_x)⁹ and
modeling simulation (Aromatics + OH + NO_x)¹⁰ are shown for comparison. It should be noted that NO_x in laboratory
experiment and modeling simulation was near 0 and we plotted them as 0.1 ppb here. The solid lines represent the logarithmic
fitting result.

**(3) The authors then use these questionable fits—based on as few as 14 datapoints, binned into four points**
**to improve the appearance of the fit—to construct an isopleth estimating OOM concentrations and GRs**
**across a wide range of AVOC and NO_x conditions. This is problematic for several reasons:**

**A. The isopleths imply predictive confidence in conditions (e.g., low NO_x and high AVOC) that are not**
**supported by the data.**

**B. Despite earlier emphasis on the importance of temperature (Figure 3c) and condensation sink, these**
**factors are omitted from the isopleth analysis.**

**C. Other critical influences, such as AVOC composition, oxidant concentrations, and RO_2 levels, are**
**ignored, flying in the face of all of the known OOM literature**

**Finally, the authors choose to present GR_{7-25} in Figure 4b, a diameter range where their simulations fail to**
**match observations (see Figure S3), undermining their conclusions. This final analysis is deeply flawed**
**and detracts significantly from the manuscript's overall quality.**

While the available OOM dataset is indeed limited, we have attempted to developed a simplified
parameterization scheme to quantify the coupled effects of concurrent AVOCs and NO_x reductions on
both condensable OOM concentration and GR in the autumn of Beijing during 2017-2021, based on OOM
data in the autumn of 2021. This parameterization scheme establishes these coupled effects through: (1)
the relationships between AVOCs and total OOM concentration, and (2) the relationships between NO_x
and f_{con} , both derived from observational data fitting. The derived condensable OOM concentration is
subsequently converted to GR using condensation potential (CP). The relationship between AVOCs and
total OOM concentration was determined through linear regression, as oxidation product concentrations
typically exhibit linear positive correlations with precursor levels. The relationship between NO_x and f_{con}
was established using logarithmic fitting because (1) the observational data exhibit a distinct logarithmic
relationship, and (2) previous laboratory experiments and modeling simulations of OOM formation from
biogenic and anthropogenic VOC oxidation consistently support this logarithmic dependence. The
detailed methodology has been revised in Methods for better understand.

“Based on the discussion above, the concurrent decline of AVOCs and NO_x since 2017 has led to a reduction in
 total OOM concentration, and an increased proportion of condensable OOMs participating in new particle growth.
 To quantify these coupled effects on condensable OOM concentration and GR during 2017-2021, we developed a
 simplified parameterization scheme based on OOM observation in 2021. The core components of this scheme
 comprise: (1) the relationships between AVOCs and total OOM concentration, and (2) the relationships between
 NO_x and f_{con} , both derived from observational data fitting in the autumn in 2021. The derived condensable OOM
 concentration is subsequently converted to GR using condensation potential (CP) (see Methods for detailed
 methodology).

Methods

Analysis on the combined effect of AVOCs and NO_x on condensable OOMs and GR

This parameterization scheme establishes the coupled effects of AVOCs and NO_x on condensable OOMs and GR
 in the autumn of Beijing during 2017-2021 through: (1) the relationships between AVOCs and total OOM
 concentration, and (2) the relationships between NO_x and f_{con} , both derived from observational data fitting in the
 autumn in 2021. The derived condensable OOM concentration is subsequently converted to GR using condensation
 potential (CP).

The relationship between AVOCs and total OOM concentration was determined through linear regression (Fig. 3a,
 $y = 2.0 \times 10^6 \times x + 2.8 \times 10^7$, $R^2 = 0.45$), as oxidation product concentrations typically exhibit linear positive
 correlations with precursor levels. The relationship between NO_x and f_{con} was established using logarithmic fitting
 (Fig. 3b). The relationship between NO_x and f_{con} for $\log_{10}(\text{C}^*) \leq -4, = -3, = -2$ are $y = -0.20 \times \log_{10}(x) + 0.84$, $R^2 =$
 0.82 ; $y = -0.19 \times \log_{10}(x) + 0.68$, $R^2 = 0.86$; and $y = -0.16 \times \log_{10}(x) + 0.53$, $R^2 = 0.88$, respectively. The logarithmic
 fitting approach was adopted based on: (1) the observational data exhibit a distinct logarithmic relationship, and (2)
 previous laboratory experiments and modeling simulations of OOM formation from biogenic and anthropogenic
 VOC oxidation consistently support this logarithmic dependence (Supplementary Fig. S23). By multiplying the
 total OOM concentration and f_{con} under different conditions of AVOCs and NO_x , the condensable OOMs
 concentration of various volatilities can be obtained. The condensable OOM concentration is the daily average value.

**Figure S23.** The relationships between daily average fraction of condensable OOMs and NO_x in the observations in 2021.
 The dashed lines represent the logarithmic fitting result. Other laboratory experiment (Monoterpene + OH/O₃ + NO_x)⁹ and
 modeling simulation (Aromatics + OH + NO_x)¹⁰ are shown for comparison. It should be noted that NO_x in laboratory
 experiment and modeling simulation was near 0 and we plotted them as 0.1 ppb here. The solid lines represent the logarithmic
 fitting result.

We employed daily CP of OOMs with different volatilities to convert daily condensable OOM concentration into
 size-resolved GR. The daily CP was calculated by linear fitting the size-resolved GR with the corresponding daily
 average condensable OOM concentration (Supplementary Fig. S24), which differs from the CP obtained based on
 OOMs concentrations during the growth period in Supplementary Fig. S9. Since condensation GR is primarily
 determined by the condensable OOM concentration during the growth period rather than the daily average
 concentration, the linear fitting between the condensation GR and daily average OOM concentration typically
 exhibits a y-intercept. When converting GR, it is necessary to consider not only the slope but also the influence of
 the intercept.

Figure S24. Condensation potential of condensable OOMs with different volatilities. Daily average vapor concentration and simulated condensation growth rate for (a) 1.5-3 nm, (b) 3-7 nm and (c) 7-25 nm particles. The vapors include sulfuric acid, OOMs with $\log_{10}(C^*) \leq -4$, $\log_{10}(C^*) = -4$, -3 and -2 . Dashed lines are the linear regressions for different vapors. (d) Slopes of linear regressions ($GR_{\text{simulated}}$ vs concentration), i.e., condensation potential (CP), for different vapors in particle size bins.

The steps for converting condensable OOM concentrations with different volatilities into size-resolved GR are as follows: First, we considered that SA concentration was constant in recent years, contributing 0.4, 0.35 and 0.25 nm h^{-1} for $GR_{1.5-3}$, GR_{3-7} and GR_{7-25} , respectively (Fig. 2). Secondly, the concentration of OOMs with $\log_{10}(C^*) \leq -4$ were substituted into their respective fitting equations with $GR_{1.5-3}$, GR_{3-7} and GR_{7-25} to obtain the contribution from OOMs with $\log_{10}(C^*) \leq -4$ to each size-resolved GR. The contributions from OOMs with $\log_{10}(C^*) \leq -4$ were then summed with those from SA to obtain the final $GR_{1.5-3}$. Next, the concentration of OOMs with $\log_{10}(C^*) = -3$ were substituted into their respective fitting equations with GR_{3-7} and GR_{7-25} to obtain the contribution from OOMs with $\log_{10}(C^*) = -3$ to each size-resolved GR. The contributions from OOMs with $\log_{10}(C^*) \leq -4$ and $\log_{10}(C^*) = -3$ were then summed with those from SA to obtain the final GR_{3-7} . Finally, the concentration of OOMs with $\log_{10}(C^*) = -2$ were substituted into their respective fitting equations with GR_{7-25} to obtain the contribution from OOMs with $\log_{10}(C^*) = -2$ to GR_{7-25} . The contributions from OOMs with $\log_{10}(C^*) \leq -4$ and $\log_{10}(C^*) = -3$ and $\log_{10}(C^*) = -2$ were then summed with those from SA to obtain the final GR_{7-25} .”

As mentioned in the comments, OOM formation are influenced by many factors apart from AVOCs and NO_x , such as temperature, condensation sink (CS), oxidant concentrations (photochemical reactivity) and AVOC composition. Here, this parameterization scheme primarily focuses on evaluating OOM formation and GR in Beijing during autumn from 2017 to 2021, where some factors can be assumed constant. For example, this scheme does not explicitly consider the effects of photochemical reactivity or CS, as both of them remained relatively stable throughout the autumn of 2017-2021. To eliminate temperature effects on volatility, the analyses were conducted using the campaign-average temperature (288 K) in the revised manuscript. The precursor source was considered relatively stable in the parameterization scheme, which could potentially affect simulation performance. The discrepancies between measurements and simulations may originate from (1) the limited data availability, and (2) the underlying assumption of stable precursor sources and other factors. Excitingly, the parameterization scheme successfully reproduced the observations during NPF days in the autumn of 2018, with discrepancies between measured and simulated condensable OOM concentration and size-resolved GR remaining below 25%. The agreement between simulations and measurements confirms that the GR increase in 2017-2021, which is primarily driven by elevated condensable OOM levels, directly results from the unbalanced emission reductions between AVOCs and NO_x . The simplified parameterization scheme may be

considered capable to quantify the coupled effects of concurrent AVOCs and NO_x reductions on both
condensable OOM concentration and GR in the autumn of Beijing during 2017-2021. As for the predictive
confidence in conditions with low NO_x and high AVOC, while our dataset cannot fully validate them,
such scenarios are atmospherically improbable and thus likely exert minimal influence on our conclusions.
In the revised manuscript, we have (1) explicitly documented all potential bias sources, (2) conducted
comparative analysis with previous studies investigating the impacts of NO_x reduction on OOM formation
and new particle growth, and (3) removed AVOCs and NO_x emission reduction scenario assessments to
ensure rigorous scientific integrity. In addition, we revised to present GR₃₋₇ in main text, a diameter range
where their simulations match observations, to strengthen our conclusions. We have made correction in
the revised manuscript as follows:

“Based on the discussion above, the concurrent decline of AVOCs and NO_x since 2017 has led to a reduction in
total OOM concentration, and an increased proportion of condensable OOMs participating in new particle growth.
To quantify these coupled effects on condensable OOM concentration and GR during 2017-2021, we developed a
simplified parameterization scheme based on OOM observation in 2021. The core components of this scheme
comprise: (1) the relationships between AVOCs and total OOM concentration, and (2) the relationships between
NO_x and f_{con} , both derived from observational data fitting in the autumn in 2021. The derived condensable OOM
concentration is subsequently converted to GR using condensation potential (CP) (see Methods for detailed
methodology). The simulation results under different AVOCs and NO_x conditions are presented in Fig. 4 and
Supplementary Fig. S19. Excitingly, the parameterization scheme successfully reproduced the observations during
NPF days in the autumn of 2018, with discrepancies between measured and simulated condensable OOM
concentration and size-resolved GR remaining below 25% (Supplementary Fig. S20). The agreement between
simulations and measurements confirms that the GR increase in 2017-2021, which is primarily driven by elevated
condensable OOM levels, directly results from the unbalanced emission reductions between AVOCs and NO_x. Note
that this scheme does not explicitly consider the effects of photochemical reactivity or CS, as both of them remained
relatively stable throughout the autumn of 2017-2021. The OOM sources were treated as stable in the scheme,
which may affect condensable OOM concentration and lead to simulation biases. Additionally, the limited
availability of OOM data in 2021 could also introduce uncertainties into the simulation results.

Discussion

The impact of NO_x on both OOMs and GR has been previously investigated in earlier studies. Li, et al.¹¹
demonstrated that high NO_x levels lead to insufficient condensable OOM concentrations and thus slow particle
growth in urban areas, which aligns well with the principal conclusions of this study. Additionally, some studies
have shown that reduced NO levels can activate nocturnal nitrogen chemistry and provide substantial sources of

chlorine radicals for daytime reactions.³⁰ This process can significantly promote the oxidation of VOCs and subsequent OOM formation. However, this enhanced atmospheric oxidation capacity may play a more critical role under polluted conditions, while its significance in new particle nucleation and growth under clean conditions may be limited.”

Fig. 4. Isopleth plots for OOMs with $\log_{10}(C^*) \leq -4$ and GR₃₋₇. Isopleth plots for (a) estimated absolute concentration of OOMs with $\log_{10}(C^*) \leq -4$ and (b) GR₃₋₇ calculated based on CP and concentration of condensable OOMs as a function of NO_x and AVOCs. Black filled circles represent the measured levels of AVOCs and NO_x in NPF days in autumn of 2018 and 2021 in Beijing. The whiskers correspond to the 25th and 75th percentiles of NO_x and AVOCs.

 **Figure S19.** Isopleth plots for OOMs with $\log_{10}(C^*) \leq -3$ and $\log_{10}(C^*) \leq -2$, and $\text{GR}_{1.5-3}$ and GR_{7-25} . (a-b) Isopleth plots for
 OOMs with $\log_{10}(C^*) \leq -3$ and $\log_{10}(C^*) \leq -2$ as a function of NO_x and AVOCs. (c-d) Isopleth plots for $\text{GR}_{1.5-3}$ and GR_{7-25} as
 a function of NO_x and AVOCs. Filled circles represent the average levels of AVOCs and NO_x in NPF days in autumn of 2018
 and 2021 in Beijing. The whiskers correspond to the 25th and 75th percentiles of NO_x and AVOCs.

**Figure S20.** The simulated and observed (a) concentrations of OOMs with different volatility ranges ($\log_{10}C^* \leq -4$, $\log_{10}C^*$
$= -3$, and $\log_{10}C^* = -2$) and (b) the GR values ($GR_{1.5-3}$, GR_{3-7} , and GR_{7-25}) in the autumn of 2018.

*2. Some dodgy averages*

*Figures 3a and 3b are based on arbitrary bins used for averaging. Extended Data Figure 8 reveals that the*
*bins for AVOCs contain very few datapoints, whereas those for NO_x include many. This discrepancy*
*undermines the robustness of the analysis, and I also question why (or even how) the authors calculated*
*the 25th and 75th percentiles for the error bars in Figure 3b. The results would be far more convincing if*
*the raw data were presented instead of relying on arbitrary binning. Additionally, the chosen bins produce*
*an alarmingly good fit, which raises concerns about the validity of the approach.*

Thanks for the comment. As mentioned in the comments, the binned methodology is unsuitable for data
analysis. We have revised all analyses and figures using raw daily average values to clearly present our
results. We sincerely apologize for the errors in the previous version of the manuscript.

In addition, we conducted the analyses about the effect of AVOCs and NO_x on OOMs using the campaign-
average temperature (289 K) in the revised manuscript to eliminate temperature effects on volatility,
which significantly improves the fitting performance.

We have made correction in the revised manuscript as follows:

Fig. 3. Impact of AVOCs and NO_x on condensable OOM concentration. **a** The relationships between daily average concentration of OOMs and AVOCs at campaign-average temperature (289 K) during the observations in 2021. The dashed line represents the linear fitting line, and the fitting result is shown in Methods. **b** The relationships between daily average fraction of condensable OOMs (f_{con}) and NO_x at average temperature (289 K) during the observations in 2021. The dashed lines represent the logarithmic fitting line, and the fitting results is shown in Methods. **c** The relationships between the concentrations of the total OOMs and temperature from different measurements. Data for spring, autumn and winter 2019 in Beijing are from Guo, et al.¹⁹, data for winter 2018 in Beijing, autumn 2018 in Nanjing and Hong Kong are from Nie, et al.¹⁴, data for autumn 2018 in Shanghai are from Tian, et al.²⁷, data for autumn 2021 in Xiamen are from Yang, et al.²⁶, and data for NPF days in autumn 2018 in Beijing are from Qiao, et al.²⁰. The area formed by the gray lines is the logarithmic fitting curve, 95% confidence and prediction band of data from the complete measurements before 2020. **d** The relationships between the average effective oxygen number (nO_{eff_ave}) of OOMs and NO_x from different measurements. Data for spring, autumn and winter 2019 in Beijing are from Guo, et al.¹⁹, data for winter 2018 in Beijing, autumn 2018 in Shanghai, Nanjing and Hong Kong are from Nie, et al.¹⁴, and data for autumn 2021 in Xiamen are from

336 Yang, et al.²⁶. The filled circles and hollow circles are average values from whole measurement and NPF days, respectively. The
337 gray error bars show standard deviations. Note that the OOMs in Xiamen is AOOMs. Given the dominant role of AOOMs in urban
atmospheres, their concentrations can serve as a reasonable proxy for all OOM levels.

3. The temperature curve

*Figure 3c: The curves presented are not sufficiently explained. It is unclear why Xiamen data were*
*excluded from the fit (is it because they only have AVOCs OOMs? If so, why is it included at all? It is*
*referred to an awful lot. Additionally, existing measurement data from Beck et al. (2021), Brean et al. (2020,*
*2024), and Kürten et al. (2016) are missing. While some datasets, such as Beck et al. (2021) ($\sim 10^7 \text{ cm}^{-3}$*
*HOM at $\sim 270 \text{ K}$) and Brean et al. (2024) ($\sim 5 \times 10^7 \text{ cm}^{-3}$ at $\sim 295 \text{ K}$), align with the curve, others, like Brean*
*et al. (2020) ($\sim 8 \times 10^7 \text{ cm}^{-3}$ at $\sim 300 \text{ K}$) and Kürten et al. (2016) ($\sim 3 \times 10^6 \text{ cm}^{-3}$ at $\sim 288 \text{ K}$), do not, although*
*my numbers here are based on an eyeballing of their graphs.*

*This section appears to argue that temperature is the primary determinant of OOM concentration, yet*
*earlier panels demonstrate a strong dependence on precursor concentrations, and earlier text highlights*
*the role of condensation sink. These inconsistencies weaken the argument, especially the final argument*
*of the paper. Furthermore, the references are incorrect, making it unclear where all datapoints originate.*
*If the Xiamen data only report AOOMs, they should not be included in the graph or the discussion.*

We sincerely appreciate the positive assessment of our research. The “OOM-temperature curve” section
has been rewritten for better understand. Our analysis showed a significant positive correlation between
total OOM concentrations and temperature in urban atmospheric observations across China. This is likely
because high temperatures (often accompanied by intense solar radiation) can lead to increased oxidant
concentrations and accelerated RO₂ auto-oxidation rates, resulting in higher OOM production (Bianchi et
al., 2019; Praske et al., 2018; Zheng et al., 2023). Additionally, high temperatures may promote elevated
biogenic emissions and elevate biological OOM concentrations, though this effect is likely less
pronounced in urban atmosphere of China. The complete observational dataset before 2020 consistently
followed a well-defined OOM-temperature curve ($R^2 > 0.9$). However, this study employed the curve
primarily to illustrate the influence of AVOC concentration. Notably, the OOM concentrations in Beijing
and Xiamen in 2021 deviated from the OOM-temperature pattern before 2020. Under comparable
temperature, the OOM concentrations in 2021 showed multiple times lower compared to 2019 and earlier
365 years. This deviation likely stems from the reduced precursor AVOC concentration in 2021, with
366 concentration of AVOC_{S_{NC} ≥ 4} in 2021 in Beijing (5.8 ppb) and Xiamen (3.4 ppb) much lower than those
367 in 2019 and earlier, such as Beijing (13.9 ppb) and Shanghai (12.2 ppb) in 2018. In addition, OOM

concentration in NPF days in 2018 and 2021 were lower than the complete observation under comparable
temperature, primarily due to the lower level of precursor AVOCs. To better highlight the effects of AVOC
emission reductions, Figure 3c has been updated to exclusively display data points from Chinese cities.

Bianchi, F. et al. Highly Oxygenated Organic Molecules (HOM) from Gas-Phase Autoxidation Involving Peroxy
Radicals: A Key Contributor to Atmospheric Aerosol. *Chemical Reviews* 119, 3472-3509,
doi:10.1021/acs.chemrev.8b00395 (2019).

Praske, E. et al. Atmospheric autoxidation is increasingly important in urban and suburban North America. 115, 64-69,
doi:10.1073/pnas.1715540115 (2018).

Zheng, P. et al. Molecular Characterization of Oxygenated Organic Molecules and Their Dominating Roles in Particle
Growth in Hong Kong. *Environmental Science & Technology* 57, 7764-7776, doi:10.1021/acs.est.2c09252 (2023).

The data in Xiamen in 2021 was included in this study because, like the Beijing observations in 2021 in
this study, they represent atmospheric conditions under lower AVOC concentration and can thus reflect
the influence of AVOC abatement on total OOM concentrations. Therefore, data in Xiamen and Beijing
in 2021 was excluded from the fitting results of observations prior to 2019. The observations in Xiamen
only reported the concentration of anthropogenic OOM (AOOMs). Based on previous observations,
AOOMs accounted for more than 50% of total OOM concentrations in urban atmosphere in China, with
the highest proportion over 80% (Nie et al., 2022; Guo et al., 2022). Thus, the AOOM concentration in
Xiamen can serve as a reasonable proxy for all OOM levels. Furthermore, it can be observed that under
the similar photochemical reactivity, the AOOM concentration in Xiamen was significantly lower than
the fitting results in 2019 and earlier years, with a difference exceeding fivefold. Therefore, even
considering the lowest proportion of AOOM in OOMs (50%), Xiamen's total OOM concentration
remained far below the fitting results, effectively demonstrating the inhibitory effect of reduced AVOC
concentrations on total OOM levels.

Nie, W. et al. Secondary organic aerosol formed by condensing anthropogenic vapours over China's megacities. *Nature*
*Geoscience* 15, 255-261, doi:10.1038/s41561-022-00922-5 (2022).

Guo, Y. et al. Seasonal variation in oxygenated organic molecules in urban Beijing and their contribution to secondary
organic aerosol. *Atmos. Chem. Phys.* 22, 10077-10097, doi:10.5194/acp-22-10077-2022 (2022).

Thank you very much for providing the extensive OOM observations outside of China. Given that this
study primarily focuses on the impact of AVOC emission reductions in China on OOM formation, these
data were not fully presented in the main text. However, we have included these results in the Supporting
Information, covering forest, remote, and urban sites in regions such as Europe, as well as most of the
findings from urban sites in China. It's found that OOMs in forest and remote environments are, in general,

higher than in urban areas. This possibly because the OOM yield of biogenic VOCs is higher than that of
anthropogenic VOCs (Berndt et al., 2016; Garmash et al., 2020). In addition, the observation results in
Leipzig during the summer of 2022 indicate that under the similar temperature, OOM concentration is
greater at the traffic site than urban background site. This also reflects the crucial role of AVOC as
precursors in OOM formation.

Berndt, T. et al. Hydroxyl radical-induced formation of highly oxidized organic compounds. *Nature Communications* 7,
13677, doi:10.1038/ncomms13677 (2016).

Garmash, O. et al. Multi-generation OH oxidation as a source for highly oxygenated organic molecules from aromatics.
*Atmos. Chem. Phys.* 20, 515-537, doi:10.5194/acp-20-515-2020 (2020).

Additionally, we have revised the references to more clearly indicate where all datapoints originate. We
have made correction in the revised manuscript as follows:

“The influence of AVOC level on OOM concentration was further corroborated by comparative analyses with other
measurements. As shown in Fig. 3c, a significant positive correlation was observed between total OOM
concentrations and temperature in urban atmospheric observations across China (Mann-Kendall test, $p < 0.05$). This
is likely because high temperatures (often accompanied by intense solar radiation) can lead to increased oxidant
concentrations and accelerated RO₂ auto-oxidation rates, resulting in higher OOM production.²³⁻²⁵ Additionally,
high temperatures may promote elevated biogenic emissions and elevate OOM concentrations, though this effect is
likely less pronounced in urban atmosphere of China. Notably, the complete observational dataset before 2020
consistently followed a well-defined OOM-temperature curve ($R^2 > 0.9$). However, results from Beijing and Xiamen
during the autumn of 2021 markedly deviated from this established curve. Under comparable temperature, the OOM
concentrations in 2021 showed multiple times lower compared to 2019 and earlier years. Since CS remained stable
in 2018-2021, this deviation likely resulted from reduced AVOC precursors in 2021. The concentration of AVOCs_{nC}
≥ 4 in 2021 in Beijing (5.8 ppb) and Xiamen (3.4 ppb) were much lower than those in 2019 and earlier, such as
Beijing (13.9 ppb) and Shanghai (12.2 ppb) in 2018.^{26,27} In addition, OOM concentration in NPF days in 2018 and
2021 were lower than the complete observation under comparable temperature, primarily due to the lower level of
precursor AVOCs. Based on the observed promoting effect of AVOCs on OOM formation, we conclude that
continued AVOC emission abatement can effectively decrease total OOM concentrations, including condensable
OOMs.”

Fig. 3. Impact of AVOCs and NO_x on condensable OOM concentration. **a** The relationships between daily average concentration of OOMs and AVOCs at campaign-average temperature (289 K) during the observations in 2021. The dashed line represents the linear fitting line, and the fitting result is shown in Methods. **b** The relationships between daily average fraction of condensable OOMs (f_{con}) and NO_x at average temperature (289 K) during the observations in 2021. The dashed lines represent the logarithmic fitting line, and the fitting results is shown in Methods. **c** The relationships between the concentrations of the total OOMs and temperature from different measurements. Data for spring, autumn and winter 2019 in Beijing are from Guo, et al.¹⁹, data for winter 2018 in Beijing, autumn 2018 in Nanjing and Hong Kong are from Nie, et al.¹⁴, data for autumn 2018 in Shanghai are from Tian, et al.²⁷, data for autumn 2021 in Xiamen are from Yang, et al.²⁶, and data for NPF days in autumn 2018 in Beijing are from Qiao, et al.²⁰. The area formed by the gray lines is the logarithmic fitting curve, 95% confidence and prediction band of data from the complete measurements before 2020. **d** The relationships between the average effective oxygen number (nO_{eff}) of OOMs and NO_x from different measurements. Data for spring, autumn and winter 2019 in Beijing are from Guo, et al.¹⁹, data for winter 2018 in Beijing, autumn 2018 in Shanghai, Nanjing and Hong Kong are from Nie, et al.¹⁴, and data for autumn 2021 in Xiamen are from

441 Yang, et al. ²⁶. The filled circles and hollow circles are average values from whole measurement and NPF days, respectively. The
442 gray error bars show standard deviations. Note that the OOMs in Xiamen is AOOMs. Given the dominant role of AOOMs in urban
atmospheres, their concentrations can serve as a reasonable proxy for all OOM levels.

**Figure S22.** The relationships between the concentrations of the total OOMs and temperature (indicates photochemical
reactivity) from different measurements. The colorful data are shown in Fig. 3c. Data for spring 2017 in Svalbard are from
Beck, et al. ⁹, data for spring 2012 in Hyytiälä are from Yan, et al. ¹⁰, data for autumn 2016 in Hyytiälä are from Zha, et al. ¹¹,
data for summer 2013 in Melpitz are from Mutzel, et al. ¹², data for summer 2018 in Barcelona are from Brean, et al. ¹³, data
for summer 2022 in Leipzig are from Brean, et al. ¹⁴. The filled circles and hollow circles are average values from whole
measurement and NPF days in urban sites, respectively. The triangle are average values from whole measurement in forest
and remote sites. The gray error bars show standard deviations. It's found that OOMs in forest and remote environments are,
in general, higher than in urban areas. This possibly because the OOM yield of biogenic VOCs is higher than that of
anthropogenic VOCs.^{15,16} In addition, the observation results in Leipzig during the summer of 2022 indicate that under the
similar photochemical reactivity, OOM concentration is greater at the traffic site than urban background site. This also reflects
the crucial role of AVOC as a precursor in OOM formation.

4. Writing quality

*The overall writing quality of this manuscript is poor, particularly in the latter sections. Numerous small*
*issues are evident, though I have only highlighted some in my specific comments. Key concepts are*
*introduced without adequate explanation—for instance, the “OOMs-temperature curve” is mentioned*
*without first establishing what it represents. Grammar and tense errors are prevalent, such as the use of*

***“OOMs-temperature curve” instead of the correct “OOM-temperature curve.” Additionally, some phrasing***
***choices are perplexing; for example, line 259 states, “Whatever, the increasing trend [...] is obvious,”***
***which is an inappropriate and unclear expression.***

***As an expert in the field, I found sections such as the discussion of figures 3c and 3d particularly dense***
***and difficult to follow. Similarly, it is unclear how figure 4 was constructed, making it challenging to***
***interpret. These sections would greatly benefit from a rewrite to present the arguments more cohesively,***
***rather than as a collection of disconnected statements. Studies such as (Jimenez et al., 2009), have***
***effectively integrated ambient mass spectral data from many studies and conveyed their findings in a clear***
***and concise manner, which this manuscript could emulate. Furthermore, the figures and captions are often***
***confusing. For example, figure captions, such as Extended Data Figure 8 deviates from the standard***
***format by failing to describe each panel sequentially, making it unnecessarily difficult to understand. The***
***word “fraction” appears on many Y axes also, but what the “fraction” is can sometimes only become***
***apparent in the figure caption. It would be much easier to state “fraction of OOMs containing nitrogen”***
***in extended data fig. 6 for example.***

Thanks for the comment. We have revised the entire text to present the arguments more cohesively, and
tried to emulate the studies such as Jimenez et al., 2009, to effectively integrated ambient OOM data from
many studies and conveyed their findings in a clear and concise manner. The concept of “OOM-temperature
curve” has been explained clearly in the revised manuscript. The inappropriate and unclear expression have
been revised to convey our finding accurately. The figures and captions have been modified to avoid
misleading. Here are some of the revised sections:

“The effective oxygen number (nO_{eff}) is a good parameter for assessing the oxidation degree of OOMs. In this study,
we observed a strong positive correlation between concentration-weighted nO_{eff} and f_{con} (Supplementary Fig. S18),
suggesting that an increase of f_{con} was accompanied by a higher oxidation degree. nO_{eff} thus can serve as a proxy
for f_{con} . Figure 3d showed a negative correlation between concentration-weighted nO_{eff} and NO_x in different
measurements, which demonstrated the inhibition effect of NO_x on condensable OOM formation from the
perspective of multi-observation comparison. For measurements conducted in Beijing, the concentration-weighted
nO_{eff} increased from 4.20 in 2019 to 4.85 in 2021 as NO_x decreased by 46%. The nO_{eff} increase was observed in
both Aro-OOMs and Ali-OOMs, with the higher proportion of OOMs with nO_{eff} above 5 in 2021 than that in 2018
and 2019 (Supplementary Fig. S13).^{14,19} Concerning specific molecules, highly abundant Ali-OOMs ($\text{C}_6\text{H}_{11}\text{NO}_6$)
and Aro-OOMs ($\text{C}_8\text{H}_{12}\text{O}_5$ and $\text{C}_8\text{H}_{11}\text{NO}_7$) in 2021 had one more effective oxygen atom than those in 2019
($\text{C}_6\text{H}_{11}\text{NO}_5$, $\text{C}_8\text{H}_{12}\text{O}_4$ and $\text{C}_8\text{H}_{11}\text{NO}_6$).¹⁹ This confirmed that AVOCs have undergone more oxidation steps under
the NO_x mitigation, which helped improving f_{con} . The OOM sources also affect f_{con} (and nO_{eff}), with the higher f_{con}

(and nO_{eff}) of Aro-OOMs than Ali-OOMs (Supplementary Fig. S13 and S17), which may account for the discrepant nO_{eff} at similar NO_x levels between different measurements. For example, measurements in Xiamen and Beijing in 2021 and Shanghai in 2018 had a higher proportion of Aro-OOM than Ali-OOMs, resulting in higher nO_{eff} than other observations under the comparable NO_x levels. Additionally, photochemical reactivity may also influence both f_{con} and O_{eff} . Together, the ongoing abatement in NO_x emission can promote the formation of highly oxidized OOMs and increased f_{con} .”

Figure S16. Impact of NO_x on nitrogen content of OOMs. (a) The relationships between daily fraction of OOMs containing nitrogen and different levels of NO_x in the observations in 2021. (b) The relationships between fraction of nitrogen-containing compounds and NO_x concentration from different measurements. Data for spring, autumn and winter 2019 in Beijing are from Guo, et al. ¹⁹, data for winter 2018 in Beijing, autumn 2018 in Shanghai and Nanjing are from Nie, et al. ¹⁴, data for autumn 2018 in Hong Kong are from Zheng, et al. ⁴⁰, data for autumn 2021 in Xiamen are from Yang, et al. ²⁵, data for spring 2013 in Hyytiälä are from Bianchi, et al. ⁴¹, data for NPF days in autumn 2018 in Beijing are from Qiao, et al. ²¹. Note that the OOMs in Xiamen is AOOMs. Given the dominant role of AOOMs in urban atmospheres, their concentrations can serve as a reasonable proxy for all OOM levels.

Reply to Reviewer 1’s specific comments (18):

1. Line 22: What does “inherent mechanism” mean?

The “inherent mechanism” refers to the mechanism driving the long-term variations in GR. To avoid the misleading, we have removed “inherent” in the revised manuscript.

“However, the long-term trend and mechanism of new particle growth in urban environment remain unclear under ongoing anthropogenic emission abatement.”

2. Line 34-35: This ending to the abstract only makes sense if you also believe that NPF is a major contributor to air quality.

According to literature reports (Kulmala et al., 2021) and our data analysis, nearly half of the particulate pollution episodes in urban Beijing originated from new particle (NPF) process, especially in spring, autumn and winter (Supplementary Fig. S3). Therefore, we believe that NPF is a major contributor to air quality in urban Beijing. Nanoparticle growth determines the survival probability of newly-formed particles to larger size in the presence of pre-existing aerosols and thus the environmental effects of NPF, especially in polluted urban atmosphere with high aerosol loading. The Supplementary Fig. S3 has been added in the revised manuscript.

Kulmala, M. et al. Is reducing new particle formation a plausible solution to mitigate particulate air pollution in Beijing and other Chinese megacities? Faraday Discussions 226, 334-347, doi:10.1039/D0FD00078G (2021).

Figure S3. Haze episodes and frequency of haze by NPF in different season from 2017 to 2021. A haze episode is defined as a continuous period with daily average PM_{2.5} concentrations exceeding 75 μg m⁻³ for two or more consecutive days.

3. Line 47: What do you mean by “speed up” haze formation? Do you mean the onset would be faster, or there would be a greater total mass during haze events? Or both?

“Speed up” means the faster haze formation. But since the eventual clearing of haze is dictated by larger-scale weather conditions in Beijing, the length of a haze episode is strongly related to how quickly haze forms. The higher growth rates of the nucleation particle mode following NPF lead to faster haze formation and longer haze episodes, which can cause the greater total mass during haze events (Kulmala et al., 2021; Kulmala et al., 2022).

Kulmala, M. et al. Is reducing new particle formation a plausible solution to mitigate particulate air pollution in Beijing

and other Chinese megacities? Faraday Discussions 226, 334-347, doi:10.1039/D0FD00078G (2021).

Kulmala, M. et al. The contribution of new particle formation and subsequent growth to haze formation. Environmental
Science: Atmospheres 2, 352-361, doi:10.1039/D1EA00096A (2022)

To avoid misleading, we have made correction in the revised manuscript as follows:

“Nanoparticle growth determines the survival probability of newly-formed particles to larger size in the presence
of pre-existing aerosols in polluted urban atmosphere,⁸ with higher growth rate (GR) accelerating haze
developments.^{5,6”}

**4. Line 51: What do you mean by certain period? Do you mean short period?**

Yes, the “Certain period” means “short period”. We have made correction in the revised manuscript as
follows:

“Although numerous NPF observations in urban areas exist,^{9,10} most short-term observations cannot establish GR
trends, hindering assessment of long-term environmental effects of NPF under ongoing anthropogenic emission
abatement.”

**5. Line 57: Where AVOCs dominate what? The VOC concentration? The reactive VOCs?**

The AVOCs dominate the VOC concentration. To avoid possible misleading, we have made correction in
the revised manuscript as follows:

“In urban atmosphere, OOMs primarily form through multi-generation oxidation and auto-oxidation of AVOCs.^{14”}

**6. Line 63: What do you mean “the long-term variation”? Do you mean “a long term variation”?**

Yes, “the long-term variation” means “a long-term variation”. We have made correction in the revised
manuscript as follows:

“In the context of continuous AVOCs and NO_x emission abatement in China since the implementation of “Three-
564 year Action Plan for Cleaner Air” in 2018,¹⁵⁻¹⁷ OOM formation may have undergone substantial changes, which
may drive a long-term variation in GR.”

**7. Line 76: I’m not sure what this sentence means “..and the volatility limits of OOMs that are able to
condense in these three particle size bins and thus contribute to GR shows distinguishable features”. What**

*are distinguishable features? Also, the figures are referenced out of order.*

The “distinguish features” mean the volatility limits of OOMs that can condense and contribute to growth
within 1.5-3 nm, 3-7 nm and 7-25 nm are different. GR_{1.5-3}, GR₃₋₇ and GR₇₋₂₅ were mainly contributed by
OOMs with $\log_{10}(C^*) \leq -4$, $\log_{10}(C^*) \leq -3$ and $\log_{10}(C^*) \leq -2$, respectively. Although we test finer
subdivisions (7-15 nm and 15-25 nm), their growth characteristics aligned with the 7-25 nm. To convey
our findings, we have made correction in the revised manuscript as follows:

“Here, three size bins (1.5-3, 3-7, and 7-25 nm) were selected for GR analysis because: (1) grown particles
typically reach ~25 nm, and (2) condensable OOMs exhibit distinct volatility limits in each bin. Although we test
finer subdivisions (7-15 nm and 15-25 nm), their growth characteristics aligned with the 7-25 nm (Supplementary
Fig. S4).”

**Figure S4.** The condensation growth rate contributed by OOMs with different volatilities for 1.5-25 nm particles. The
volatility distribution of condensation GR in 7-15 nm and 15-25 nm are similar with that in 7-25 nm

**8. Line 105 and following:** you need to explain what data you used here. You start the prior section saying
“Long-term measurements of particle number size distribution (PNSD)...”.

Thanks for the useful comment. We have explained the data we used in the beginning of this section.

“To understand the cause of the long-term increase in size-resolved GR, we first identified the main contributors
for new particle growth using CI-APi-ToF measurements in the autumn of 2021 in urban Beijing.”

**9. Line 152:** Is it EXACTLY the same period? Or is it just two autumn time datasets?

It is not the same period in the autumn of 2018 and 2021, it is two autumn time datasets. We have made
correction in the revised manuscript as follows:

“Here, we compared the volatility distribution of OOMs in NPF days in the autumn of 2021 with that reported in
the autumn of 2018 (Fig. 2b-c).”

**10. Line 177: Please explain the nO_{eff} here.**

Thanks for the comment. We have made correction in the revised manuscript as follows:

“(d) The relationships between the average effective oxygen number (nO_{eff}) of OOMs and NO_x from different
measurements.”

**11. Line 183: Due, rather than duo?**

Thanks for the comment. We have made correction in the revised manuscript as follows:

“Gaseous condensable OOMs can easily condense onto pre-existing particles and be removed due to their low
volatility.”

**12. Line 197: This claim is a bit sudden. What is the “OOM-temperature curve”? I understand this in
principle, but it is not sufficiently explained.**

Thanks for the comment. We have made correction in the revised manuscript as follows:

“The influence of AVOC level on OOM concentration was further corroborated by comparative analyses with other
measurements. As shown in Fig. 3c, a significant positive correlation was observed between total OOM
concentrations and temperature in different observation in urban China (Mann-Kendall test, $p < 0.05$). This is likely
because high temperatures (often accompanied by intense solar radiation) can lead to increased oxidant
concentrations and accelerated RO_2 auto-oxidation rates, resulting in higher OOM production.²³⁻²⁵ Additionally,
high temperatures may promote elevated biogenic emissions and elevate OOM concentrations, though this effect is
likely less pronounced in urban China. Notably, the complete observational dataset before 2020 consistently
followed a well-defined OOM-temperature curve ($R^2 > 0.9$). However, results from Beijing and Xiamen during the
autumn of 2021 markedly deviated from this established curve. Under comparable temperature, the OOM
concentrations in 2021 showed multiple times lower compared to 2019 and earlier years. Since CS remained stable
in 2018-2021, this deviation likely resulted from reduced AVOC precursors in 2021. The concentration of $AVOC_{S_{nC}}$

≥ 4 in 2021 in Beijing (5.8 ppb) and Xiamen (3.4 ppb) were much lower than those in 2019 and earlier, such as Beijing (13.9 ppb) and Shanghai (12.2 ppb) in 2018.^{26,27} In addition, OOM concentration in NPF days in 2018 and 2021 were lower than the complete observation under comparable temperature, primarily due to the lower level of precursor AVOCs. Based on the observed promoting effect of AVOCs on OOM formation, we conclude that continued AVOC emission abatement can effectively decrease total OOM concentrations, including condensable OOMs.”

Fig. 3. Impact of AVOCs and NO_x on condensable OOM concentration. **a** The relationships between daily average concentration of OOMs and AVOCs at campaign-average temperature (289 K) during the observations in 2021. The dashed line represents the linear fitting line, and the fitting result is shown in Methods. **b** The relationships between daily average fraction of condensable OOMs (f_{con}) and NO_x at average temperature (289 K) during the observations in 2021. The dashed lines represent the logarithmic fitting line, and the fitting results is shown in Methods. **c** The relationships between the concentrations of the total OOMs and temperature from different measurements. Data for spring, autumn and winter 2019 in Beijing are from Guo, et al.¹⁹, data for winter

2018 in Beijing, autumn 2018 in Nanjing and Hong Kong are from Nie, et al.¹⁴, data for autumn 2018 in Shanghai are from Tian, et al.²⁷, data for autumn 2021 in Xiamen are from Yang, et al.²⁶, and data for NPF days in autumn 2018 in Beijing are from Qiao, et al.²⁰. The area formed by the gray lines is the logarithmic fitting curve, 95% confidence and prediction band of data from the complete measurements before 2020. **d** The relationships between the average effective oxygen number (nO_{eff}) of OOMs and NO_x from different measurements. Data for spring, autumn and winter 2019 in Beijing are from Guo, et al.¹⁹, data for winter 2018 in Beijing, autumn 2018 in Shanghai, Nanjing and Hong Kong are from Nie, et al.¹⁴, and data for autumn 2021 in Xiamen are from Yang, et al.²⁶. The filled circles and hollow circles are average values from whole measurement and NPF days, respectively. The gray error bars show standard deviations. Note that the OOMs in Xiamen is AOOMs. Given the dominant role of AOOMs in urban atmospheres, their concentrations can serve as a reasonable proxy for all OOM levels.

13. Fig 3d: Why are certain sites (non-chinese cities) excluded from this analysis?

Thanks for the comment. This study focuses on the atmospheric conditions in Chinese urban areas, specifically analyzing the impacts of China's emission-reduction policies on OOM formation and GR. Thus, our analysis centers on observational data from Chinese cities. Additionally, non-Chinese urban sites provided limited data on concentration-weighted O_{eff} . Therefore, we primarily present results from Chinese cities in Fig. 3d.

14. Line 206: Multi-generation oxidation OR auto-oxidation

Thanks for the comment. We have made correction in the revised manuscript as follows:

“Most condensable OOMs are highly oxidized products generated by multi-generation oxidation or auto-oxidation of RO_2 radicals.”²⁶”

II. Reply to Reviewer 3

Reply to Reviewer 3's overall comments:

*Tang et al. report the long-term trend of size-resolved particle growth rate in urban area of Beijing during*
*the autumns from 2017 to 2021, and explore the driving factor of growth rate variations based on an intensive*
*campaign with the gaseous oxygenated organic molecules observations. These long-term data can be an*
*important reference for future air pollution control and mitigation works in China. The manuscript is well-*
*written and the presentation is clear. However, I have some queries on the novelty and discussion in the*
*manuscript.*

We appreciate the constructive comments from the reviewer on this manuscript. We have answered them
point to point in the following paragraphs (the texts italicized are the comments, the texts indented are the
responses, and the texts in blue are revised parts in new manuscript). And we have carefully modified the
language before submitting. In addition, all changes made are marked in the revised manuscript.

Reply to Reviewer 3's main concerns (3):

*1. I think that the manuscript should be significantly revised to emphasize more on its novelty and include*
*more solid discussion to show the importance of the long-term observations from a broader perspective.*
*For example, the authors stated in Line 65-67 that relevant ambient measurement evidence is still scarce,*
*and the mechanism of new particle growth and its long-term evolution need in-depth research. I do agree*
*that the long-term observations of GR are important, however, I do not see that the overall results in the*
*current manuscript have addressed this statement well enough. There are already some previous studies*
*that focus on the long-term observations of new particles growth related to the OOMs in urban Beijing (e.g.,*
*Li et al., 2022, ES&T; Guo et al., 2023, ACP). Other than different in time period (season), how do the*
*authors distinguish the novelty of this study compared to the previous observations? This should be clarified.*
*Furthermore, it is not mentioned in the manuscript on why only the autumn data were selected for the*
*analysis. Or why it is important to look at the autumn data? Any justification?*

Thanks for the insightful comment. We have recognized that some expressions in our original manuscript
may have caused ambiguity. This study does not present long-term interannual variations of OOMs, due
to limitations in instrument availability, while certain phrasing might have inadvertently implied
otherwise. We have carefully revised these sections to eliminate any potential ambiguity.

The novelty of this work lies in: we established and quantified the relationships between precursor
concentrations (including AVOCs and NO_x) and both condensable OOMs and growth rates (GR) using
available OOM data, and assessing and interpreting the changes in OOMs formation and GR under

ongoing anthropogenic emission reductions based on long-term variations in precursors — without
requiring interannual OOM measurements. Several previous studies have reported long-term OOM
observations, but they primarily focused on seasonal variations or correlations with GR. Few studies have
systematically examined how sustained emission reductions affect OOM evolution and GR. Considering
the significant impact of new particle growth on urban haze pollution, our work provides important
scientific support for air quality improvement strategies. We have emphasized the novelty include more
solid discussion to show the importance of the long-term observations in the revised manuscript. The
following shows the main modifications:

“Although numerous NPF observations in urban areas exist,^{9,10} most short-term observations cannot establish GR
trends, hindering assessment of long-term environmental effects of NPF under ongoing anthropogenic emission
abatement. Gaseous sulfuric acid (SA) drives nucleation and initial growth (< 3 nm), and oxygenated organic
molecules (OOMs) condensation has been proposed to dominate subsequent growth to sub-micro particles.¹¹⁻¹³ In
urban atmosphere, OOMs primarily form through multi-generation oxidation and auto-oxidation of AVOCs.¹⁴
However, high NO_x level strongly perturb oxidation process by terminating the RO₂ radical auto-oxidation, and
thus increase OOM volatility.^{12,14} In the context of continuous AVOCs and NO_x emission abatement in China since
the implementation of Air Pollution Prevention and Control Action Plan (2013–2017) and the Three-year Blue-sky
Action Plan (2018–2020),¹⁵⁻¹⁷ OOM formation may have undergone substantial changes, which may drive a long-
term variation in GR. Recent studies indicated that the shift in OOM formation under continuous air quality
improvement might be beneficial for new particle growth.^{11,17,18} However, long-term OOM measurement evidence
remain scarce due to limited mass spectrometry availability. While some extended OOM observation exist, they
mainly addressed seasonal variations or GR correlations, rather than interannual evolution during air quality
improvements.^{11,19} Although long-term OOM precursors data (e.g., VOCs and NO_x) are available, but
parameterization schemes linking these to OOMs and GR are lacking. Developing such schemes would advance
understanding of OOM formation, new particle growth mechanisms, and environmental impacts under ongoing
anthropogenic emission abatement — particularly in the absence of long-term OOM or GR observations.

In this work, we conducted comprehensive measurements in Beijing, a Chinese megacity, to investigate the trends
and mechanisms of new particle growth. The frequent NPF events in Beijing substantially affect air quality and
public health, making the study of new particle growth in Beijing crucial.²⁰ At the same time, Beijing has
implemented long-term atmospheric emission control measures, making it an ideal environment to investigate how
such mitigation measures affect new particle growth. We discovered the increasing trends in urban Beijing during
autumn from 2017 to 2021. Based on the quantitative relationship originated from comprehensive measurements of
gaseous oxygenated organic molecules (OOMs) in autumn 2021, we have identified the primary driver for annual

GR variations. We first employed a dynamic condensation model with OOM data in 2021 to identify the main
contributors for new particle growth. Comparative analysis against data in 2018 confirmed that the observed GR
enhancement stemmed from increased condensable OOM concentrations. Then, we qualitatively recognized the
coupled effect of AVOCs and NO_x on condensable OOMs concentration based on OOM data in 2021 and
comparative analysis with previous OOM studies. Finally, we established a parameterization scheme linking AVOC
and NO_x concentrations to condensable OOM level and size-resolved GR, to validate the critical role of concurrent
AVOC and NO_x emission abatement in long-term variations of both OOMs and GR. Our study highlights the critical
importance of targeted anthropogenic emission controls for effective air quality improvement.

**Discussion**

In this study, we observed increasing trends of size-resolved GR in urban Beijing during autumn from 2017 to 2021.
Based on a quantitative relationship originated from comprehensive measurements of gaseous oxygenated organic
molecules (OOMs) in autumn 2021, we have demonstrated that the unbalanced emission abatement between
AVOCs and NO_x were the primarily driver of the increased GR. While AVOCs abatement decreased total OOM
concentration, the concurrent NO_x reduction led to an increase in the fraction and concentration of highly oxidized
condensable OOMs that can participate in new particle growth, thereby resulting in increased GR.

These findings emphasize the need for comprehensive long-term precursor observations to fully elucidate the
dynamic evolution and fundamental mechanisms of atmospheric particle growth, which is critical for developing
targeted pollution control strategies.”

Autumn was selected for this study due to its high frequency of haze episodes, particularly those triggered
by new particle formation (NPF). As shown in Supplementary Fig. S3, the frequency of haze episodes
and haze driven by NPF process in Beijing during the autumn from 2017 to 2021 is higher than other
seasons. This suggests that the environmental impacts of NPF are more pronounced during autumn, which
is why our study primarily focuses on the autumn results in Beijing. To explain our seasonal selection
criteria, we have made correction in the revised manuscript as follows:

“Particle number size distribution (PNSD, 1.5-700 nm) and trace gases (NO_x, SO₂, O₃, and AVOCs) were measured
in urban Beijing during autumn (September to November) from 2017 to 2021, when haze episodes, particularly
those triggered by NPF are most frequent (Supplementary Fig. S3).”

Figure S3. Haze episodes and frequency of haze by NPF in different season from 2017 to 2021. A haze episode is defined as a continuous period with daily average $PM_{2.5}$ concentrations exceeding $75 \mu g m^{-3}$ for two or more consecutive days.

2. In term of mechanism, linking the growth rate with the OOMs evolution and NO_x level is a key point of this study. It is unfortunate that the OOMs data were only available in 2021, and the trend of the OOMs levels were unknown in other years. Although the authors have implied that the AVOCs decrease will cause decrement in total OOMs (and condensable OOMs) and compared the data from previous studies to show the impacts of NO_x to highly oxidized OOMs formation (Fig.3), how sure are they that trend will not deviate as predicted? It is stated that the ongoing abatement in NO_x emission promoted the formation of highly oxidized OOMs and increased fcon (Line 236-237). Despite supported by the calculations, I am not totally convinced on this conclusion by the current data as they could be also influenced by other factors as shown by previous studies. Yan et al. (2022, ACP) found that OOM composition and volatility were insensitive to the large change of atmospheric NO_x concentration in Beijing during the COVID-19 restrictions; instead, the associated high particle growth rates and high OOM concentration during the lockdown period were mostly caused by the enhanced atmospheric oxidative capacity, which linked to the enhanced night-time NO_x chemistry (Yan et al., 2023, NG). Furthermore, Li et al. (2022, EST) exhibits a different view, where they found that the insufficient condensable organic vapours lead to slow growth, which further causes low survival of the newly formed particles in urban environments of Beijing. Did the author compare their results with more recent findings to get a bigger picture on the trend of OOMs? For example, Yuan et al. (2024, ES&T) reported the atmospheric gaseous OOMs in urban Beijing from January 2022 and January 2023. Can these data being used to support the reduction of NO_x leading to OOMs decrease beyond 2021?

Thanks for the comment. This study proposes that, the NO_x emission reduction led to an increase in the fraction and concentration of highly oxidized condensable OOMs involved in new particle growth in

Beijing, thereby resulting in an increase in GR during autumn from 2017 to 2021. We first discovered the
increasing trends in urban Beijing during autumn from 2017 to 2021. We then employed a dynamic
condensation model with OOM data in 2021 to identify the main contributors for new particle growth.
We compared the OOM observation in the autumn 2021 with that in autumn 2018, confirming that the
observed increase in GR originates from elevated concentrations of condensable OOMs. Next, based on
OOM data in autumn 2021 (including the introduction of f_{con}), and comparative analysis with previous
observation data (incorporating metrics such as the total OOM concentration and effective oxygen number
(nO_{eff})), we qualitatively recognized the coupled effect of AVOCs and NO_x on condensable OOMs
concentration. Finally, we attempted to establish a parameterization scheme linking AVOC and NO_x
concentrations with condensable OOMs and size-resolved GR, to validate the critical role of concurrent
AVOC and NO_x emission abatement in long-term variations of both OOMs and GR. This scheme
successfully reproduces the OOM observations in NPF days in 2018. The agreement between simulations
and measurements confirms that the GR increase in 2017-2021, which is primarily driven by elevated
condensable OOM levels, directly results from the unbalanced emission reductions between AVOCs and
NO_x . All the above analyses have focused on explaining the changes in OOMs and GR during autumn
2017-2021. During this period, photochemical reactivity and condensation sink (CS) remained relatively
stable, and thus would not affect our conclusions.

As mentioned in the comment, several previous studies have examined the effects of NO_x on OOM
formation and GR. Yan et al. (2022, ACP) found that OOM composition and volatility were insensitive
to the large change of atmospheric NO_x concentration in Beijing during the COVID-19 restrictions. The
insensitivity of OOM composition and volatility may stem from differences in photochemical reactivity
between the Pre-Lockdown and Lockdown periods. These two periods (December-January and January-
March, respectively) exhibit distinct seasonal characteristics. For instance, temperatures during the
Lockdown period were approximately 6 K higher than during Pre-Lockdown, with O_3 concentrations 25%
greater. The observed differences in OOMs between these periods may not be entirely attributable to NO_x
effects. Previous studies have shown that elevated temperatures coupled with reduced NO_x concentrations
lead to increased RO_2 production, thereby consequently diminishing the influence of NO_x on RO_2
chemistry. Our work specifically examines OOM and GR trends during autumn (2017-2021) in Beijing,
a period characterized by stable photochemical reactivity and condensation sink (CS) conditions. This
environmental stability minimizes confounding effects from external variables such as temperature,
thereby providing more robust attribution of precursor AVOCs and NO_x impacts on OOM formation and
GR.

Yan, C. *et al.* The effect of COVID-19 restrictions on atmospheric new particle formation in Beijing. *Atmos. Chem. Phys.* **22**, 12207-12220, doi:10.5194/acp-22-12207-2022 (2022).

Furthermore, Yan et al. (2023, NG) proposed the associated high particle growth rates and high OOM concentration during the lockdown period were mostly caused by the enhanced atmospheric oxidative capacity, which linked to the enhanced night-time NO_x chemistry. This study showed that reduced NO levels can activate nocturnal nitrogen chemistry and provide substantial sources of chlorine radicals for daytime reactions. This process can significantly promote the oxidation of VOCs and subsequent OOM formation. However, this enhanced atmospheric oxidation capacity may play a more critical role under polluted conditions ($> 150 \mu\text{g m}^{-3}$), while its significance in new particle nucleation and growth under clean conditions may be limited (Fig. R1). Therefore, we consider this process to be less influential for new particle growth, despite its significant role in secondary aerosol formation.

Figure R1. Relative contribution of VOC oxidation by different atmospheric oxidants, including Cl radicals, OH radicals, NO_3 radicals and O_3 (Yan et al., 2023, NG)

Yan, C. *et al.* Increasing contribution of nighttime nitrogen chemistry to wintertime haze formation in Beijing observed during COVID-19 lockdowns. *Nature Geoscience* **16**, 975-981, doi:10.1038/s41561-023-01285-1 (2023).

Li et al. (2022, EST) demonstrated that high NO_x levels lead to insufficient condensable OOM concentrations and thus slow particle growth in urban areas, which aligns well with the principal conclusions of this study. However, this study only qualitatively demonstrated the influence of NO_x on the proportion of OOMs with $\log_{10}\text{C}^* \leq -1$, without exploring the combined effects of AVOCs and NO_x ,

despite acknowledging the importance of such interactions. Our work provides an integrated qualitative-
quantitative assessment of the synergistic effects between AVOCs and NO_x on critical condensable OOMs,
effectively addressing knowledge gaps in previous research.

Li, X. *et al.* Insufficient Condensable Organic Vapors Lead to Slow Growth of New Particles in an Urban Environment.
*Environmental Science & Technology* **56**, 9936-9946, doi:10.1021/acs.est.2c01566 (2022).

Based on these considerations, we have added comparative discussions of previous studies in the revised
manuscript to present our conclusions in a more scientifically systematic manner.

“The impact of NO_x on both OOMs and GR has been previously investigated in earlier studies. Li, et al.¹¹
demonstrated that high NO_x levels lead to insufficient condensable OOM concentrations and thus slow particle
growth in urban areas, which aligns well with the principal conclusions of this study. Additionally, some studies
have shown that reduced NO levels can activate nocturnal nitrogen chemistry and provide substantial sources of
chlorine radicals for daytime reactions.³⁰ This process can significantly promote the oxidation of VOCs and
subsequent OOM formation. However, this enhanced atmospheric oxidation capacity may play a more critical role
under polluted conditions, while its significance in new particle nucleation and growth under clean conditions may
be limited.”

Thanks for your advice for utilizing the recent finding to get a bigger picture on the trend of OOMs. Yuan
et al. (2024, ES&T) reported the atmospheric gaseous OOMs in urban Beijing from January 2022 and
January 2023. However, we found that this study utilizing CI-Orbitrap measure OOMs, which differs
from the CI-APi-TOF used in our analysis and most observations. As mentioned in the paper, there are
distinctions between CI-Orbitrap and CI-APi-TOF in the identification of OOM molecules, which may
impact the final OOM results. For example, the study reported that the proportion of nitrogen-containing
OOMs in Beijing across four seasons was approximately 80%, which is higher than those in previous
studies in urban Chinese (65%-80%, Extended Data Fig. 7). Therefore, this data may not be applicable to
our study. Additionally, we have not found more recent OOM observational data measured by CI-APi-
TOF in urban China. In light of this, our analysis primarily focuses on the evolution of OOMs and GR
under AVOC and NO_x abatement during the autumn from 2017 to 2021. Base on the identified critical
roles of AVOCs and NO_x in condensable OOM formation, this study suggested that priority control of
AVOCs (especially aromatic VOCs) may be an effective strategy for GR decreasing and air-quality
improvement. In addition, we expanded the discussion on other factors that may influence OOM
formation or new particle growth processes, and emphasizing the importance of long-term OOM
observations in the Policy implications.

Figure S7. Impact of NO_x on nitrogen content of OOMs. (a) The relationships between daily fraction of OOMs containing nitrogen and different levels of NO_x in the observations in 2021. (b) The relationships between fraction of nitrogen-containing compounds and NO_x concentration from different measurements. Data for spring, autumn and winter 2019 in Beijing are from Guo, et al.¹⁹, data for winter 2018 in Beijing, autumn 2018 in Shanghai and Nanjing are from Nie, et al.¹⁴, data for autumn 2018 in Hong Kong are from Zheng, et al.⁴⁰, data for autumn 2021 in Xiamen are from Yang, et al.²⁵, data for spring 2013 in Hyytiälä are from Bianchi, et al.⁴¹, data for NPF days in autumn 2018 in Beijing are from Qiao, et al.²¹. Note that the OOMs in Xiamen is AOOMs. Given the dominant role of AOOMs in urban atmospheres, their concentrations can serve as a reasonable proxy for all OOM levels.

“Previous studies have demonstrated that elevated GR exacerbate haze formation and severity, highlighting GR reduction as critical for pollution mitigation.^{5,6} Our analysis reveals that prioritizing AVOCs control over NO_x may more effectively suppress GR. The source-segregated OOM contribution is a vital basis for targeted AVOCs controlment. Simulation showed that anthropogenic OOMs dominated the total OOM contribution to GR in all size ranges (~88%; Supplementary Fig. S21), with Aro-OOMs being the predominant contributor (~72% of total OOMs) due to their high concentration and f_{con} . Ali-OOMs represented the second largest source (~16%), while biogenic OOMs exhibited minimal contribution to GR. These findings suggest that priority control of aromatic VOCs may be an effective strategy for GR reduction and air quality improvement.

Beyond direct effects on OOM formation, precursor VOCs and NO_x may also indirectly influence GR by altering oxidant levels (e.g., O₃ and chlorine radicals), which should be considered in future research. Furthermore, iodine oxoacids and sulfuric acids have been proven to contribute to the particle growth (particularly for sub-3nm particles) in polluted urban environments, making it another crucial factor to consider in future GR mitigation strategies. These findings emphasize the need for comprehensive long-term precursor observations to fully elucidate the dynamic evolution and fundamental mechanisms of atmospheric particle growth, which is critical for developing targeted pollution control strategies.”

***3. I agree that developing effective strategies to reduce GR is important for mitigating aerosol pollution,***
***but the current discussion in the policy implication section is simplistic. It is not sure how the calculation***
***led to the proposal of 1:3 AVOCs and NO_x abatement ratio. I am wondering if the changes of 1:3 to 1:2***
***will have a significant effect in the OOMs concentration considering all the uncertainties in the calculation***
***(including the uncertainty in AVOCs and NO_x measurements)? Maybe I am wrong, but at least from***
***arrows in the Fig. 4 and EDF 9, they seem not to show significant different (large shift) in concentrations***
***of OOMs to me. As I have mentioned in the comment above, many other factors can affect the OOMs level,***
***but the limited OOMs observation data (only in 2021) has weakened the validity of the conclusion drawn***
***in this section. I think that the effectiveness to reduce GR in urban area is more complicated than just by***
***controlling the AVOC (aromatic VOCs) because besides the VOCs levels, the levels of oxidants may also***
***play important role in the OOMs formation. This has been neglected throughout the discussion in the text.***
***In addition, iodine oxoacids and sulfuric acids have been proven to contribute to the particle formation and***
***particle growth in polluted urban environments (e.g., Zhang et al., 2024, ACP), may further complicate the***
***control measure in reducing particle growth. I would suggest the authors to revise the implication of this***
***study.***

Thanks for the comment. As mentioned in the comments, OOM formation are influenced by many factors
apart from AVOCs and NO_x, such as temperature, condensation sink (CS), oxidant concentrations
(photochemical reactivity). Therefore, our analysis, particularly the parameterization scheme for
condensable OOMs, primarily focuses on evaluating OOM formation and GR in Beijing during autumn
from 2017 to 2021, where some factors such as photochemical reactivity (indicated by O_x levels and
temperature) or CS can be assumed constant. The parameterization scheme may not remain applicable for
future implication if other factors change. Thus, we removed AVOCs and NO_x emission reduction
scenario assessments as suggested, to ensure rigorous scientific integrity. In addition, we fully
acknowledge that the the effectiveness to reduce GR in urban area is complicated besides by controlling
the AVOC (aromatic VOCs). Thus, we expanded the discussion on other factors that may influence OOM
formation or new particle growth processes, such as oxidant concentrations, iodine oxoacids and sulfuric
acids in Policy implications. We have made correction in the revised manuscript as follows:

“Previous studies have demonstrated that elevated GR exacerbate haze formation and severity, highlighting GR
reduction as critical for pollution mitigation.^{5,6} Our analysis reveals that prioritizing AVOCs control over NO_x may
more effectively suppress GR. The source-segregated OOM contribution is a vital basis for targeted AVOCs
controlment. Simulation showed that anthropogenic OOMs dominated the total OOM contribution to GR in all size

ranges (~88%; Supplementary Fig. S21), with Aro-OOMs being the predominant contributor (~72% of total OOMs)
due to their high concentration and f_{con} . Ali-OOMs represented the second largest source (~16%), while biogenic
OOMs exhibited minimal contribution to GR. These findings suggest that priority control of aromatic VOCs may
be an effective strategy for GR reduction and air quality improvement.

Beyond direct effects on OOM formation, precursor VOCs and NO_x may also indirectly influence GR by altering
oxidant levels (e.g., O_3 and chlorine radicals), which should be considered in future research. Furthermore, iodine
oxoacids and sulfuric acids have been proven to contribute to the particle growth (particularly for sub-3nm particles)
in polluted urban environments, making it another crucial factor to consider in future GR mitigation strategies.
These findings emphasize the need for comprehensive long-term precursor observations to fully elucidate the
dynamic evolution and fundamental mechanisms of atmospheric particle growth, which is critical for developing
targeted pollution control strategies.”

**Reply to Reviewer 3’s specific comments (16):**

***1. Line 1: The title of the manuscript is overstated to me, suggest to revise the title accordingly. For example,***
***“Urban China” is not suitable here as I think the current data in the manuscript only represent urban***
***Beijing, instead of the general urban areas of China. I did not see sufficient data to support the general***
***urban areas of China.***

Thanks for the comment. Considering that the data in this study primarily represent urban Beijing, we
have revised “urban China” to “a Chinese megacity” in the revised manuscript as follows:

“Ongoing anthropogenic emission abatement promotes new particle growth in a Chinese megacity”

***2. Line 45: Should be environmental effects.***

Tanks for the comment. We have made correction in the revised manuscript.

***3. Line 76: What kind of distinguishable features?***

The “distinguish features” mean the volatility limits of OOMs that can condense and contribute to growth
within 1.5-3 nm, 3-7 nm and 7-25 nm are different. $\text{GR}_{1.5-3}$, GR_{3-7} and GR_{7-25} were mainly contributed by
OOMs with $\log_{10}(\text{C}^*) \leq -4$, $\log_{10}(\text{C}^*) \leq -3$ and $\log_{10}(\text{C}^*) \leq -2$, respectively. Although we test finer
subdivisions (7-15 nm and 15-25 nm), their growth characteristics aligned with the 7-25 nm. To convey
our findings, we have made correction in the revised manuscript as follows:

“Here, three size bins (1.5-3, 3-7, and 7-25 nm) were selected for GR analysis because: (1) grown particles
typically reach ~25 nm, and (2) condensable OOMs exhibit distinct volatility limits in each bin. Although we test
finer subdivisions (7-15 nm and 15-25 nm), their growth characteristics aligned with the 7-25 nm (Supplementary
Fig. S4).”

**Figure S4.** The condensation growth rate contributed by OOMs with different volatilities for 1.5-25 nm particles. The
volatility distribution of condensation GR in 7-15 nm and 15-25 nm are similar with that in 7-25 nm

**4. Line 77-78: Please revise the sentence as it is hard to understand how the measured GR and trace gases**
**can directly link to the anthropogenic emission abatement and changing atmospheric chemistry?**

Thanks for the comment. We have made correction in the revised manuscript as follows:

“Given the meteorological stability in 2017-2021 (Supplementary Fig. S5), its impact on atmospheric pollutant
concentrations is negligible. Thus, the observed GR and trace gas variations primarily reflect anthropogenic
emission changes and atmospheric chemistry.”

**5. Line 87: Please state the method for estimating the condensational sink (CS) data in the methodology or**
**supporting information.**

Thanks for the reviewer’s comment. The method for estimating the condensational sink (CS) data have
been added in supporting information.

“To evaluate the scavenging effects of preexisting particles on condensable vapors, the condensation sink (CS) was
calculated as follow:²

$$CS = 2\pi D \sum \beta_m(D_{p,i}) D_{p,i} N_i$$

(2)

where D is the diffusion coefficient of the condensing vapor, β_m is the transition regime correction factor, and
$D_{p,i}$ and N_i are the diameter and number concentration in the size class i , respectively.

Dal Maso, M., Kulmala, M., Riipinen, I. & Wagner, R. Formation and growth of fresh atmospheric aerosols: Eight
963 years of aerosol size distribution data from SMEAR II, Hyytiälä, Finland. *Boreal Environment Research* **10**, 323-
964 336 (2005).”

**6. Line 89: Suggest rewording ‘in’ to ‘throughout’.**

Thanks for the comment. We have made correction in the revised manuscript.

**7. Line 91: Suggest to include the standard deviation for the averaged value of growth rate.**

Thanks for the comment. We have made correction in the revised manuscript as follows:

“The averaged GR in 3-7 nm and 7-25 nm (GR_{3-7} and GR_{7-25}) increased from 2.2 ± 1.2 and 2.9 ± 1.3 nm h⁻¹ in
2017 to 3.2 ± 1.3 and 3.9 ± 1.7 nm h⁻¹ in 2021, with the annual increasing rates of 0.15 to 0.38 nm h⁻¹ yr⁻¹ and 0.12
to 0.32 nm h⁻¹ yr⁻¹, respectively.”

**8. Line 131: accompanied by?**

Thanks for the comment. We have made correction in the revised manuscript.

**9. Line 148: The volatility distribution of OOMs in 2018 were obtained from Qiao et al., are the OOMs**
**detected in Qiao’s study similar to the current study? If not, what are the differences?**

Since we lack molecular-level OOM data for 2018, a direct molecular comparison cannot be made.
However, based on the information provided by Qiao et al., the proportion of nitrogen-containing
components in OOMs during NPF days in our study appears lower than their reported results (70% vs
76%). Furthermore, when comparing with the mass defect plots of OOMs in spring in Qiao et al., our
OOM data demonstrate a higher oxygenation degree. Specifically, our study shows higher concentrations
of $C_xH_yNO_{6-8}$ and $C_xH_yN_2O_{8-10}$ species, whereas Qiao et al. reported higher concentrations of $C_xH_yNO_{5-6}$
and $C_xH_yN_2O_{7-8}$ compounds (Fig. R2). Although the compared seasons differ, considering that Beijing’s

spring typically has lower NO_x levels and higher photochemical reactivity than autumn, we anticipate that
the composition of OOMs in autumn 2018 Beijing likely exhibited a lower oxidation degree than during
spring.

**Figure R2.** Mass defect plot of the OOMs identified in NPF days in the autumn of 2021 (left, this study) and spring of 2018
(right, Qiao et al.). The dots connected by lines represent the homologs or serial products.

Qiao, X. et al. Contribution of Atmospheric Oxygenated Organic Compounds to Particle Growth in an Urban
Environment. *Environmental Science & Technology* 55, 13646-13656, doi:10.1021/acs.est.1c02095 (2021).

**10. Line 259: Please change ‘Whatever’ to a more appropriate word (same for line 289).**

Thanks for the comment. We have made correction in the revised manuscript.

**11. Line 304: Should be ‘i.e.’,**

000 Thanks for the comment. We have made correction in the revised manuscript.

001

002 **12. For the sulfuric acid calibration, the authors should state the detail procedures of their calibration**
003 **conducted for this study instead of just mentioning the work of Kürten et al. A calibration curve of H₂SO₄**
004 **should be included to the supporting information as this is important because the concentration of OOMs**
005 **were also based on this calibration. What is the determined limit of detection and zeroing background signal**
006 **for H₂SO₄? All these information should also be added.**

007 Thanks for the comment. The detailed information for the sulfuric acid calibration has been added in
008 Method and supporting information.

009 “The calibration was conducted in the sampling line, and the diffusion loss was contained in the calibration result.

The calibration curve of H₂SO₄ is shown in Supplementary Fig. S1. The calibration coefficients of H₂SO₄ is $1.23 \times 10^{10} \text{ cm}^{-3}$ in the autumn of 2021. The zeroing background signal for H₂SO₄ measured with the CI-APi-TOF is $5.0 \times 10^4 \text{ molec cm}^{-3}$ (1 min integration). The limit of detection (LOD) for H₂SO₄ is $9.0 \times 10^4 \text{ molec cm}^{-3}$ and it is defined as three times the standard deviation of the background.”

Figure S1. Time series of a typical H₂SO₄ calibration experiment and the calibration curve retrieved from the above time series. f62, f125, f188, f97 and f160 are the signal of NO₃⁻, HNO₃·NO₃⁻, (HNO₃)₂·NO₃⁻, HSO₄⁻ and H₂SO₄·NO₃⁻.

13. Line 387: Revise the typo.

Thanks for the comment. We have made correction in the revised manuscript.

14. Line 662, Extended Data Fig. 1: Why the AVOCs comparison plot (figure b) only included autumn 2018 and 2021 data, while the VOCs were also measured in autumn 2017 (as mentioned in the methodology)? Please justify. Does it still have a decrease trend after including the 2017 data? In addition, it is not mentioned that what kind/specific species of AVOCs were used for the analysis in this study. This information should be added to a table in the supporting information.

Thanks for the comment. The AVOCs data in the autumn of 2017 has been added in Extended Data Fig.

1. The decreased trend is still observed after including data in 2017. The species of AVOCs used for analysis has been added in supporting information.

Figure S6. Variations of NO_x and AVOCs on NPF days and non-event days in the autumn from 2017 to 2021. (a) Variations of NO_x on NPF days and non-event days. (b) Variations of AVOCs, AVOCs with nC ≥ 4 as well as aromatic and aliphatic VOCs in NPF days and non-event days in the autumn in 2017, 2018 and 2021. The whiskers correspond to the 25th and 75th percentiles. The circular and cross markers represent the median and mean values, respectively.

Table S3. AVOC species used for the analysis in this study.

Type	Species	nC	Type	Species	nC
	Ethane	2		Benzene	6
	Ethylene	2		Toluene	7
	Propane	3		Ethylbenzene	8
	Propylene	3		m/p-Xylene	8
	Isobutane	4		o-Xylene	8
	Trans-2-butene	4		Styrene	8
Aliphatic	1-Butene	4	Aromatic	iso-Propylbenzene	9
	Cis-2-butene	4		n-Propylbenzene	9
	Cyclopentane	5		m-ethyltoluene	9
	Isopentane	5		p-ethyltoluene	9
	n-Pentane	5		1,3,5-Trimethylbenzene	9
	1-Pentene	5		o-ethyltoluene	9
	trans-2-Pentene	5		1,2,4-Trimethylbenzene	9
Aliphatic	cis-2-Pentene	5		1,2,3-Trimethylbenzene	9

2,3-Dimethylbutane	6	m-diethylbenzene	10
2-Methylpentane	6	p-diethylbenzene	10
3-Methylpentane	6		
1-Hexene	6		
n-Hexane	6		
Methylcyclopentane	6		
Cyclohexane	6		
2-Methylhexane	7		
2,3-Dimethylpentane	7		
3-Methylhexane	7		
n-Heptane	7		
Methylcyclohexane	7		
2,2,4-Trimethylpentane	8		
2,3,4-Trimethylpentane	8		
2-Methylheptane	8		
3-Methylheptane	8		
n-octane	8		
n-Nonane	9		
n-Decane	10		

035

036 **15. Line Extended Data Fig.4: It is not sure why the Hyytiala (a forest station) data was included for the**
 037 **comparison of OOMs to data from cities in China. It confuses me as the current manuscript is discussing**
 038 **the OOMs in urban China, isn't it?**

039 Thanks for the comment. As mentioned in the comments, this study focuses on the atmospheric conditions
 040 in Chinese urban areas. We have removed the data in Hyytiala in the figure.

041

042 **Figure S12. The concentration of the total OOMs and OOMs originated from different precursors. (a) Comparison of the total**

OOMs (red), Aro-OOMs (blue) and Ali-OOMs (green) at different locations in autumn and winter. Data for autumn 2021 in Xiamen are from Yang, et al. ²⁵, data for autumn 2019 in Beijing are from Guo, et al. ¹⁹, data for winter 2018 in Beijing, autumn 2018 in Shanghai, Nanjing and Hong Kong are from Nie, et al. ¹⁴, and data for NPF days in autumn 2018 in Beijing are from Qiao, et al. ²¹. The pink point represents the OOM concentration observed in NPF days. Points are mean concentrations, and bars on x-axis direction correspond to the 25th and 75th values. (b) Absolute concentrations and contributions of OOMs originated from different precursors in NPF days. Aro-, Ali-, MT-, IP-OOMs and Undis stands for OOMs originated from aromatics, aliphatics, monoterpene, isoprene and undistinguished source.

16. Supporting Information, Figure S9: Suggest to separate the r^2 value according the linear lines as the range of 0.59-0.80 is quite big. It is better to show which line has stronger correlation.

Thanks for the comment. We have separated the R^2 value according the linear lines.

Figure S18. Relationship between the fraction of condensable OOMs (f_{con}) and concentration-weighted nO_{eff} of OOMs in the observations in 2021. The dash lines are the linear regressions of f_{con} and concentration-weighted nO_{eff} of OOMs.

Reply to comments

Manuscript ID: NCOMMS-24-78518A

Title: “Ongoing anthropogenic emission abatement promotes new particle growth in a Chinese megacity”

Author(s): Lizi Tang, Zeyu Feng, Dongjie Shang, Linghan Zeng, Zhijun Wu, Hui Wang, Shiyi Chen, Xin Li, Limin Zeng, Jianlin Hu, Min Hu

I. Reply to Reviewer 1

Reply to Reviewer 1’s overall comments:

The authors have generally done a good job of responding to my comments. The analysis of the HOM data is first class and they’ve convinced me that the isopleth figure is worth keeping, with one caveat, which is that I think the fact that there is no evidence that their model works outside of the parameter space of their measurements needs to be noted.

We sincerely appreciate the positive assessment and the constructive comments from the reviewer on this manuscript. We have answered them point to point in the following paragraphs (the texts italicized are the comments, the texts indented are the responses, and the texts in blue are revised parts in new manuscript). And we have carefully modified the language before submitting. In addition, all changes made are marked in the revised manuscript.

Reply to Reviewer 1’s general concerns (2):

1. The analysis of the HOM data is first class and they’ve convinced me that the isopleth figure is worth keeping, with one caveat, which is that I think the fact that there is no evidence that their model works outside of the parameter space of their measurements needs to be noted.

Thanks for the comment. It is true that we have no evidence to suggest that our model remains valid outside of the parameter space of the measurements, such as the conditions with low NO_x and high AVOCs. To avoid the possible misleading, caveats around the generalisability of the isopleth model has be added in the revised manuscript:

“We emphasize that this parameterization scheme was derived from measurements in the autumn of 2021, and its
validity outside the observed parameter space remains unconfirmed, requiring further verification in future research.”

**2. Figures one through three present some really nice analysis. I feel like the authors felt some pressure to**
**tie it together into some coherent story and this has led to some obfuscations and exaggerations throughout**
**the manuscript. Similarly, the English throughout is still below a publishable standard. I’ll take one snippet**
**(at random) as an example**

**“Gaseous condensable OOMs can easily condense onto pre-existing particles and be removed due to their**
**low volatility (1). The CS remained stable throughout both the whole days and NPF days since 2017 (2),**
**suggesting that the increase in condensable OOM concentration were independent of CS variations (3). In**
**urban atmosphere, OOM formation primarily related to photochemical oxidation of AVOCs (~80%,**
**Supplementary Fig. S10 and S12). The relatively constant photochemical reactivity (indicated by Ox levels**
**and temperature) during 2017-2021 exerted negligible influence on condensable OOM concentration**
**variations. (4)”**

**1) “condense onto pre-existing particles and be removed” is ambiguous as it could imply that the**
**condensation and the removal are two separate processes. They instead of “condense onto pre-existing**
**particles, resulting in their removal” or something similar**

**2) “whole days” is incorrect, I think they mean “The CS remained stable both across all measurement days**
**and across NPF days”.**

**3) Should say “Condensable OOM concentrations”**

**4) Ox levels and temperature do not indicate “photochemical reactivity”. Similarly, “2017-2021” is**
**misleading here because it implies that they know how photochemical reactivity affected OOMs from 2017**
**through 2021. In reality they do not have a measure of photochemical reactivity, and if they did, they only**
**have OOM measurements for a small fraction of the 2017 to 2021 period.**

**It is outside of the scope of a reviewer’s job to highlight and fix every single one of these as they’re consistent**
**through the manuscript.**

Thanks for the comment. We have carefully revised the expression for technical correctness and to ensure
claims do not go beyond what the data can evidence. And the English language has been carefully
improved throughout.

Some revisions are shown as follows:

“The increased concentrations of condensable OOMs since 2017 could result from variations in their precursor
sources, oxidation processes, and condensation sinks. Our analysis shows that approximately 80% of OOMs formed
through AVOC oxidation (Supplementary Fig. S10 and S12), suggesting that declining AVOC emissions since
2017 likely drive the reduction in OOM concentrations. The response of OOM concentrations to AVOCs will be
discussed later based on the intensive campaign in autumn 2021 and comparison with other measurements.
Temperature and oxidant concentrations are two essential factors affecting OOM oxidation processes.^{17,24}
Observational results demonstrated that both temperature and O_x (as a proxy for oxidant levels) remained relatively
stable both across all measurement days and across NPF days since 2017 (Supplementary Fig. S5), indicating that
neither temperature nor oxidant concentrations were the key factors controlling the increase in condensable OOM
concentrations. Additionally, NO_x can influence OOM oxidation processes by terminating RO₂ radical auto-
oxidation in urban atmospheres.²⁵ The sustained NO_x emission reduction since 2017 may have affected OOM
composition and condensable OOM concentrations, as will be discussed later. The CS remained stable both across
all measurement days and across NPF days since 2017 (Supplementary Fig. S5), suggesting that the increase in
condensable OOM concentrations was independent of sink variations.”

“The influence of AVOC levels on OOM concentrations was further corroborated through comparative analyses
with other measurements. As shown in Fig. 3c, significant positive correlations were observed between total OOM
concentrations and temperature in urban atmospheric observations across China (Mann-Kendall test, $p < 0.05$). This
likely occurs because high temperature (often accompanied by intense solar radiation) increase oxidant
concentrations and accelerated RO₂ auto-oxidation rates, thereby enhancing OOM production.^{24,26,27} Additionally,
high temperature may promote elevated biogenic emissions and increase biological OOM concentrations, although
this effect is likely less pronounced in urban atmospheres in China.²⁶ Notably, the average results from
measurements prior to 2020 followed a well-defined OOM-temperature curve ($R^2 > 0.9$). However, results from
Beijing and Xiamen during autumn 2021 markedly deviated from this established curve. Under comparable
temperature, OOM concentrations in 2021 were multiple times lower than those in 2019 and earlier years. Potential
drivers for the observed deviation include difference in precursor sources or CS. Observation data revealed that the
CS levels in Beijing and Xiamen during 2021 were comparable to or lower than those in previous measurements,
suggesting that the lower AVOC levels in 2021 may represent the primary explanatory factor. The campaign-average
concentrations of AVOC_{S_{NC} ≥ 4} in 2021 in Beijing (5.8 ppb) and Xiamen (3.4 ppb) were significantly lower than
those in 2019 and earlier years (e.g., 13.9 ppb in Beijing and 12.2 ppb in Shanghai during 2018).^{28,29} In addition,
the reduced AVOC precursor levels led to decreased OOM concentrations in NPF days than those in all measurement
89 days under comparable temperature in autumn 2018 and 2021. Based on the observed promoting effect of AVOCs
on OOM formation, we conclude that the continued AVOC emission abatement would effectively decrease total

OOM concentrations, including condensable OOMs.”

“Based on the preceding discussion, the concurrent decline in AVOCs and NO_x since 2017 have yielded two
competing effects: (1) an overall decrease in total OOM concentrations, and (2) an increased proportion of
condensable OOMs participating in particle growth. However, the coupled effects from AVOC and NO_x reduction
and their roles in driving the long-term increase in condensable OOM concentrations and GR remain unclear. To
address these questions, we developed a simplified parameterization scheme based on the OOM observation in
autumn 2021, specifically quantifying the response of condensable OOM concentrations and GR to anthropogenic
emission abatement during autumn from 2017 to 2021. The core components of this scheme comprise: (1) the
relationships between AVOC concentrations and total OOM concentrations, and (2) the relationships between NO_x
concentrations and f_{con} , both derived from observational data fitting during autumn 2021. The derived condensable
OOM concentrations are then converted to GR using the condensation potential (CP) (see Methods for details). The
simulation results under different AVOC and NO_x conditions are presented in Fig. 4 and Supplementary Fig. S19.
Notably, the parameterization scheme successfully reproduced the observations during NPF days in autumn 2018,
with remaining discrepancies between measured and simulated condensable OOM concentrations and size-resolved
GR below 25% (Supplementary Fig. S20). The successful application of the parameterization scheme (developed
using the dataset in 2021) to the dataset of 2018 provides critical evidence that the observed increase in condensable
OOMs and GR during 2017-2021 was primarily driven by the predominance of NO_x abatement’s enhancing effects
over AVOC abatement’s inhibitory effects. Note that this scheme does not explicitly consider the effects of
temperature, oxidant concentrations, or CS, as these parameters remained relatively stable throughout 2017-2021.
The OOM sources were treated as unchanged in the scheme, which may introduce simulation biases. Furthermore,
the limited availability of OOM data in 2021 could also introduce additional uncertainties. We emphasize that this
parameterization scheme was derived from measurements in autumn 2021, and its validity outside the observed
parameter space remains unconfirmed, requiring further verification in future research.”

**3. One final comment is that, despite my previous comment, it’s still not totally clear to me what data is**
**used to construct which plot. The new Figure 3a and b should show the same thing (daily averages), yet**
**one has 14 datapoints while the other has 26.**

Thanks for the comment. Both Figure 3a and 3b show the daily average data from the observation period
in autumn, 2021. Due to maintenance work on the online gas chromatography and mass spectrometry
(GC-MS) system, the AVOC data is relatively limited with 14 datapoints. In contrast, the NO_x data is
more complete with more datapoints. Our analysis indicated that the f_{con} -NO_x relationships for days with
available AVOC data are largely consistent with those derived from all measurement days (Figure S25).

It suggested that the key findings are robust although the AVOC and NO_x datapoints do not fully align
(Figure 3a and 3b). To avoid the possible misleading, the relevant explanations have been added in the
revised manuscript:

“Text S2. Supplementary instruments

Tracer gaseous pollutants were continuously detected by a series of online monitoring system manufactured by
Thermo Electron Corporation (O₃ (Model 49i) and NO-NO₂-NO_x (model 42i-TLE)). 99 types of volatile organic
compounds (VOCs) were measured by the online gas chromatography and mass spectrometry (GC-MS) system.⁶
The AVOC species used for the analysis in this study are shown in Table. S3. Due to maintenance work on GC-MS
system, the AVOC data is relatively limited. In contrast, the NO_x data is more complete. Our analysis indicated that
the f_{con}-NO_x relationships for days with available AVOC data are largely consistent with those derived from all
measurement days (Figure S25). It suggested that the key findings are robust although the AVOC and NO_x
datapoints do not fully align (Fig. 3a and 3b).”

**Figure S25.** The relationships between daily average fraction of condensable OOMs (f_{con}) and NO_x at average temperature (289 K)
for days with available AVOC data and lack of AVOC data, respectively. The dashed lines represent the logarithmic fitting line of
datapoints from all measurement days (the same as Fig. 3b).

**4. I'd be extremely happy to see a final version of the manuscript where the English has been tied up.**

We sincerely appreciate the positive assessment of our research. We have tried to modify the English
language throughout to improve readability in the revised manuscripts.

Some revisions are shown as follows:

“Introduction

New particle growth is a complex process involving multiple vapors. Gaseous sulfuric acids (SA) are important for
nucleation and initial growth (< 3 nm), while condensation of oxygenated organic molecules (OOMs) has been
proposed to dominate subsequent growth into sub-micro particles.⁹⁻¹¹ In urban atmospheres, OOMs primarily form
through multi-generation oxidation and auto-oxidation of anthropogenic volatile organic compounds (AVOCs).¹²
However, high NO_x levels strongly perturb the oxidation process by terminating the RO₂ radical auto-oxidation,
consequently reducing oxidation degree and increasing volatility of OOMs.^{10,12} With the progressive
implementation of the Air Pollution Prevention and Control Action Plan (2013–2017) and the Three-year Blue-sky
Action Plan (2018–2020) in China,¹³⁻¹⁵ substantial reductions in AVOC and NO_x emissions have potentially altered
OOM formation process, which may influence long-term trends of GR. Nevertheless, our understanding on the
long-term evolution of new particle growth and its governing mechanisms under continuous emission controls
remains incomplete. Observational data revealed that GR of particles in 3-25 nm exhibited no expected decline
during winter from 2013 to 2019 in Beijing, despite continuous reduction in gaseous precursors.¹⁵ Some studies
suggest that the stable GR may be attributed to NO_x reduction promoting the formation of low-volatility condensable
OOMs that are directly relevant for new particle growth, thus potentially offset the suppression effects from
precursor source controls.^{9,15,16} However, this hypothesis lacks sufficiently evidence due to the limitation of long-
term OOM measurements, despite existing comprehensive precursor concentration data (e.g., AVOCs and NO_x).
Existing extended OOM observations have mainly focused on seasonal variations or GR correlations, leaving their
interannual evolution and potential impact on new particle growth during ongoing air quality improvement poorly
understood.^{9,17}

In this work, we conducted comprehensive measurements in urban Beijing, a megacity in China, to investigate the
long-term trends and mechanisms of new particle growth. Beijing is a typical city to study new particle growth
under anthropogenic mitigation because of its frequent NPF processes and long-term emission control measures.^{15,18}
We used ambient observations to discover the GR trends during autumn from 2017 to 2021. We identified the effects
of AVOCs and NO_x on OOMs formation through Nitrate-CI-APi-ToF measurements during autumn 2021 combined
with comparative analyses with previous measurements. Importantly, we validated the critical role of concurrent
precursor (AVOC and NO_x) emission abatement in the long-term variations of both condensable OOM
concentrations and GR, using a parameterization scheme derived from the measurements during autumn 2021. Our
results underscore the significant impact of anthropogenic emission abatement on new particle growth,
demonstrating that strategic emission controls are crucial for effectively reducing GR and improving air quality.”

II. Reply to Reviewer 3

Reply to Reviewer 3's overall comments:

*I have read the authors response to my comments and those of the other reviewer and find that they have done*
*a good job in addressing the comments. I have only two minor suggestions as below.*

We sincerely appreciate the positive assessment and the constructive comments from the reviewer on this
manuscript. We have answered them point to point in the following paragraphs (the texts italicized are the
comments, the texts indented are the responses, and the texts in blue are revised parts in new manuscript). And
we have carefully modified the language before submitting. In addition, all changes made are marked in the
revised manuscript.

Reply to Reviewer 3's specific comments (2):

*1. Line 274-276: References should be provided to the works that have proven the iodine oxoacids and*
*sulfuric acids contributions to particle growth.*

Thanks for the comment. We have added the references in the revised manuscript as follows:

“Furthermore, iodine oxoacids and sulfuric acids have been proven to contribute to the particle growth (particularly
for sub-3nm particles) in polluted urban environments,³² making it another crucial factor to consider in future GR
mitigation strategies.”

³² Zhang, Y. *et al.* Iodine oxoacids and their roles in sub-3 nm particle growth in polluted urban environments.
*Atmos. Chem. Phys.* **24**, 1873-1893, doi:10.5194/acp-24-1873-2024 (2024).

*2. Fig.S1 in the supplement: The authors should describe the H₂O (lpm) in the caption. Non-expert reader*
*will not understand what does H₂O (lpm) represent for and how do they work.*

Thanks for the comment. The detailed information for the H₂O (lpm) has been added in the caption of
Fig. S1 as follows:

Figure S1. Time series of a typical H₂SO₄ calibration experiment and the calibration curve retrieved from the above time series. The information of calibration experiment can be found in Kürten, et al.¹² H₂SO₄ is produced from the reaction of SO₂ and OH radicals, which is produced in situ through the UV photolysis of H₂O. Therefore, different flow rate of H₂O (H₂O [μpm]) correspond to different H₂SO₄ concentration. f62, f125, f188, f97 and f160 are the signal of NO₃⁻, HNO₃·NO₃⁻, (HNO₃)₂·NO₃⁻, HSO₄⁻ and H₂SO₄·NO₃⁻, respectively.

12 Kürten, A., Rondo, L., Ehrhart, S. & Curtius, J. Calibration of a chemical ionization mass spectrometer for the measurement of gaseous sulfuric acid. *The journal of physical chemistry. A* **116**, 6375-6386, doi:10.1021/jp212123n (2012).